# HIF-1 Interacts with TRIM28 and DNA-PK to release paused RNA polymerase II and activate target gene transcription in response to hypoxia

Yongkang Yang[1,2], Haiquan Lu[1,2], Chelsey Chen[1,3], Yajing Lyu [1], Robert N. Cole [4] &
Gregg L. Semenza [1,2,4,5✉]

Hypoxia-inducible factor-1 (HIF-1) is a transcription factor that acts as a regulator of oxygen ($O_2$) homeostasis in metazoan species by binding to hypoxia response elements (HREs) and activating the transcription of hundreds of genes in response to reduced $O_2$ availability. RNA polymerase II (Pol II) initiates transcription of many HIF target genes under non-hypoxic conditions but pauses after approximately 30–60 nucleotides and requires HIF-1 binding for release. Here we report that in hypoxic breast cancer cells, HIF-1 recruits TRIM28 and DNA-dependent protein kinase (DNA-PK) to HREs to release paused Pol II. We show that HIF-1α and TRIM28 assemble the catalytically-active DNA-PK heterotrimer, which phosphorylates TRIM28 at serine-824, enabling recruitment of CDK9, which phosphorylates serine-2 of the Pol II large subunit C-terminal domain as well as the negative elongation factor to release paused Pol II, thereby stimulating productive transcriptional elongation. Our studies reveal a molecular mechanism by which HIF-1 stimulates gene transcription and reveal that the anticancer effects of drugs targeting DNA-PK in breast cancer may be due in part to their inhibition of HIF-dependent transcription.

[1] Institute for Cell Engineering, Johns Hopkins University School of Medicine, Baltimore, MD, USA. [2] Sidney Kimmel Comprehensive Cancer Center at Johns Hopkins, Baltimore, MD, USA. [3] Johns Hopkins University, Baltimore, MD, USA. [4] Department of Biological Chemistry, Johns Hopkins University School of Medicine, Baltimore, MD, USA. [5] McKusick-Nathans Department of Genetic Medicine, Johns Hopkins University School of Medicine, Baltimore, MD 21205, USA. ✉email: gsemenza@jhmi.edu

Decreased $O_2$ availability (hypoxia) is a key feature of the solid tumor microenvironment and a major driving force for cancer progression[1,2]. Hypoxia increases the stability and activity of the hypoxia-inducible factors (HIFs), which are heterodimeric transcription factors that are composed of an $O_2$-regulated HIF-α subunit (HIF-1α, HIF-2α, or HIF-3α) and constitutively expressed HIF-1β subunit[3,4]. HIFs induce the expression of hundreds of genes, including many that enable cells to decrease $O_2$ consumption, by switching from oxidative to glycolytic metabolism, or increase $O_2$ delivery, by stimulating angiogenesis[5,6]. Under normoxic conditions, the HIF-α subunits are subjected to $O_2$-dependent prolyl hydroxylation and are bound by the von Hippel-Lindau (VHL) protein, which recruits an E3 ubiquitin-protein ligase complex that targets HIF-α subunits for proteasomal degradation[3,7]. Under hypoxic conditions, however, hydroxylation is inhibited, HIF-α subunits are stabilized and translocate to the nucleus, where they dimerize with HIF-1β, bind to the sequence 5′-(A/G)CGTG-3′ within hypoxia response elements (HREs) of target genes[6], and recruit histone modifying enzymes[8–10], histone readers[11], chromatin remodeling proteins[12], and Mediator proteins[13] to stimulate target gene transcription by RNA polymerase II (Pol II).

Pol II assembles a pre-initiation complex consisting of transcription factors IIA, IIB, IID, IIE, IIH, and Core Mediator[14]. Cyclin-dependent kinase 7 (CDK7) and its allosteric activator Cyclin H are components of TFIIH, and CDK7 phosphorylates serine-5 (S5) in a heptad repeat (YSPTSPS) present in 52 copies in the C-terminal domain (CTD) of the large subunit of Pol II, leading to recruitment of the histone methyltransferase SETD1, which trimethylates histone H3 on lysine-4 to form the H3K4me3 mark that is associated with transcription initiation[15,16]. Although it is conventional wisdom that transcriptional activators stimulate Pol II recruitment and transcription initiation, recent studies indicate that Pol II binds to the promoter and initiates transcription but pauses approximately 30–60 nucleotides downstream of the transcription start site, as a result of its interaction with negative elongation factor (NELF) and DRB-sensitivity inducing factor (DSIF). This pausing is the rate-limiting step in transcription of 40–70% of genes induced by a physiological stimulus, such as heat shock or hypoxia[16–18]. In response to an inducing stimulus, positive transcription elongation factor b (P-TEFb), consisting of CDK9 and cyclin T1, is recruited to paused Pol II through a variety of mechanisms and CDK9 phosphorylates three critical targets: NELF, causing it to be released from Pol II; the SPT5 subunit of DSIF, converting it into a positive elongation factor; and serine-2 (S2) in the Pol II CTD, which allows it to engage positive elongation factors, such as SPT6, and chromatin-modifying enzymes, such as SETD2, which generates the H3K36me3 mark that is associated with productive transcriptional elongation[16–19].

HIF-1 was shown to recruit CDK9 to target genes in response to hypoxia, and transcriptional elongation of a subset of target genes was dependent on recruitment of CDK8, but the target(s) of CDK8 that are critical for elongation were not identified[13]. In the case of heat shock, tripartite motif containing protein 28 (TRIM28) was shown to play a critical role in mediating both Pol II pausing and release after heat shock, which resulted in phosphorylation of TRIM28 on S824 by the kinase ataxia telangiectasia mutated (ATM) or DNA-dependent protein kinase (DNA-PK), and inhibitors of these kinases blocked transcriptional elongation[18]. However, the mechanisms underlying the induction of ATM and DNA-PK activity in response to heat shock remain undetermined, although it appears to involve DNA damage[19].

In the present study, we uncovered a critical role for TRIM28 and DNA-PK in HIF-1-mediated transcription in hypoxic human breast cancer cells. HIF-1 recruits TRIM28 to HREs by direct interaction and the two proteins mediate assembly of the DNA-PK catalytic subunit (DNA-PKcs) and its regulatory subunits KU70 and KU86 into a catalytically active complex, which phosphorylates TRIM28 at S824, enabling CDK9 recruitment. RNA sequencing of knockdown subclones revealed that hypoxic induction of the vast majority of HIF-regulated genes in human breast cancer cells is dependent on TRIM28 and DNA-PK.

## Results

**HIF-1α recruits TRIM28 and DNA-PK to HREs of hypoxia-inducible genes.** To identify transcriptional co-factors that co-localize with HIF-1 in chromatin of breast cancer cells, we employed RIME (rapid immunoprecipitation and mass spectrometry of endogenous proteins), which has proven to be an efficient and unbiased proteomic approach to discover transcriptional regulators[20,21]. MDA-MB-231 human breast cancer cells were treated with the prolyl hydroxylase inhibitor and HIF-inducer dimethyloxalylglycine (DMOG; 1 mM) or vehicle control (DMSO) for 8 h, followed by protein/DNA crosslinking in 1% formaldehyde. Nuclei were isolated and sonicated, followed by immunoprecipitation with HIF-1α antibody (Fig. 1A). A total of 1081 unique proteins that co-purified with HIF-1α were identified by mass spectrometry. The identified proteins included known HIF-1α-interacting proteins, such as HIF-1β, HSP90, PARP1, PKM, RUVBL1/PONTIN, and RUVBL2/REPTIN[22,23]. However, a single sample was analyzed by RIME for each treatment condition, such that validation of individual putative interacting proteins is required to definitively establish co-localization with HIF-1α.

Four proteins (TRIM28 and the DNA-PK subunits DNA-PKcs/PRKDC, KU70/XRCC6 and KU86/XRCC5), which showed RIME spectrum counts and fold enrichment in the DMOG sample that were comparable to HIF-1β (Fig. 1B), were considered relatively high confidence hits of biological interest. To determine whether TRIM28 or DNA-PK was recruited to the HREs of HIF target genes under hypoxic conditions, we exposed MDA-MB-231 cells to 20% or 1% $O_2$ and performed chromatin immunoprecipitation (ChIP) and quantitative real-time PCR (qPCR) using antibody against TRIM28, KU70, KU86, DNA-PKcs or HIF-1α. The results showed that, similar to HIF-1α, TRIM28 and all three subunits of DNA-PK were recruited to the HREs of two prototypical HIF target genes, *ANGPTL4* (HRE is located in the 5′-flanking region, 2012 base pairs upstream of the transcription start site)[24] and *PDK1* (HRE is located in the first intron, 760 base pairs downstream of the transcription start site)[25], in response to hypoxia, whereas no recruitment was observed at the *RPL13A* gene, which is not hypoxia-induced or HIF-regulated (Fig. 1C). Hypoxia-induced occupancy of the *ANGPTL4* and *PDK1* HREs by TRIM28, KU70, KU86, DNA-PKcs, and HIF-1α was also observed in SUM159 human breast cancer cells (Supplementary Fig. 1A). Hypoxia-induced occupancy of HREs of 8 additional HIF target genes (*LOX, LDHA, CA9, VEGFA, PKM2, SLC2A3, PGK1,* and *PGF*) by TRIM28 and DNA-PK subunits was also observed (Supplementary Fig. 1B). These data demonstrate that hypoxia induces occupancy by TRIM28 and DNA-PK of HIF target gene HREs in human breast cancer cells.

We next investigated whether HIF-1α interacts with TRIM28 and DNA-PK in the absence of chromatin through co-immunoprecipitation (co-IP) assays in MDA-MB-231 cells. Endogenous TRIM28, DNA-PKcs, KU70 and KU86 were specifically co-immunoprecipitated by HIF-1α antibody (but not IgG) from whole cell lysate (WCL) of cells exposed to 1% $O_2$, but not from WCL of cells exposed to 20% $O_2$ (Fig. 1D). The interaction of TRIM28 with HIF-1α was further confirmed by the

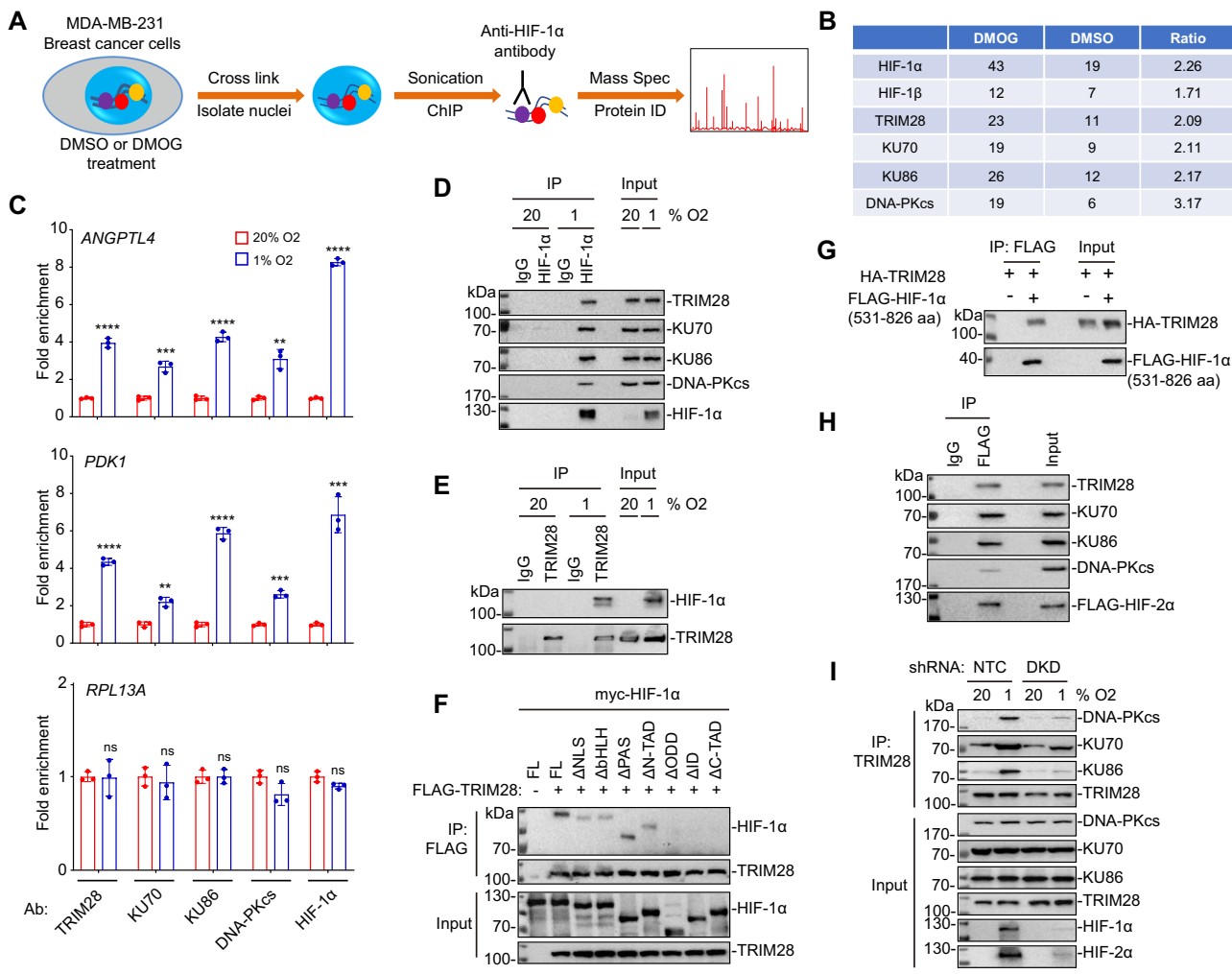

**Fig. 1 HIF-1α recruits TRIM28 and DNA-PK to HREs. A** Proteins interacting with HIF-1α in chromatin were identified by RIME. **B** For each protein, the number of peptides identified in the DMOG- (HIF inducer) and DMSO- (vehicle) treated samples, and the DMOG/DMSO ratio are shown. **C** MDA-MB-231 cells were exposed to 20% or 1% $O_2$ for 16 h, and ChIP assays were performed using the indicated antibody (Ab). Primers encompassing HRE sites in the *ANGPTL4* or *PDK1* gene were used for qPCR and *RPL13A* was analyzed as a negative control. For each primer pair, the qPCR data were normalized to the mean result at 20% $O_2$. Values were expressed as the mean ± SEM of three biologically independent samples. **p < 0.01, ***p < 0.001, ****p < 0.0001; ns, not significant (p > 0.05) by two-tailed Student's t-test. Exact p values from left to right: <0.0001, 0.0007, <0.0001, 0.0023, <0.0001 (ANGPTL4); <0.0001, 0.0019, <0.0001, 0.0002, 0.0005 (PDK1); 0.9285, 0.6396, 0.9815, 0.072, 0.054 (RPL13A). **D, E** Cells were exposed to 20% or 1% $O_2$ for 8 h. Immunoprecipitation (IP) was performed using HIF-1α (**D**) or TRIM28 (**E**) Ab or IgG, followed by immunoblot assays. **F** FLAG-TRIM28 and individual myc-tagged HIF-1α deletion mutants were co-expressed in MDA-MB-231 cells. IP with FLAG Ab was performed, followed by immunoblot assays. **G** MDA-MB-231 cells were transfected with expression vector(s) encoding FLAG-HIF-1α (531–826) and/or HA-TRIM28. IP was performed using FLAG Ab or IgG followed by immunoblot assays. **H** Cells were transfected with FLAG-HIF-2α expression vector and exposed to 1% $O_2$ for 8 h. IP was performed using FLAG Ab or IgG followed by immunoblot assays. **I** MDA-MB-231 cells were stably transfected with lentiviral vectors encoding a non-targeting control (NTC) short hairpin RNA (shRNA) or shRNAs targeting HIF-1α and HIF-2α (double knockdown, DKD). The cells were exposed to 20% or 1% $O_2$ for 8 h and IP was performed with TRIM28 Ab, followed by immunoblot assays.

inverse co-IP using an antibody against TRIM28 (Fig. 1E). To determine which domains of HIF-1α were responsible for binding to TRIM28, we co-transfected MDA-MB-231 cells with expression vectors encoding FLAG-tagged TRIM28 and one of a series of myc-epitope-tagged HIF-1α deletion mutants (Supplementary Fig. 1C). The co-IP results indicated that deletion of HIF-1α amino acid residues 401–603 (ΔODD), 575–786 (ΔID) or 786–826 (ΔC-TAD) abrogated its interaction with TRIM28 (Fig. 1F). Further investigation revealed that expression of FLAG-tagged HIF-1α residues 531–826 was sufficient for interaction with HA-tagged TRIM28 (Fig. 1G). Co-expression of the myc-tagged HIF-1α deletion constructs and FLAG-GFP-tagged KU86 revealed that deletion of HIF-1α residues 575–786

or 786–826 abrogated interaction with KU86 (Supplementary Fig. 1D).

To investigate whether TRIM28 and DNA-PK also interact with HIF-2α in response to hypoxia, MDA-MB-231 cells were transfected with a FLAG-HIF-2α expression vector, exposed to 1% $O_2$ for 8 h, and WCLs were prepared for co-IP assay. FLAG-HIF-2α interacted with TRIM28, KU70, KU86, and DNA-PKcs (Fig. 1H). Most importantly, analysis of MDA-MB-231 subclones expressing either a non-targeting control shRNA (NTC) or shRNAs targeting HIF-1α and HIF-2α (double-knockdown, DKD) revealed hypoxia-inducible association of TRIM28 with DNA-PK subunits in a HIF-dependent manner (Fig. 1I). To further investigate whether recruitment of TRIM28 and DNA-PKcs to HREs is HIF-

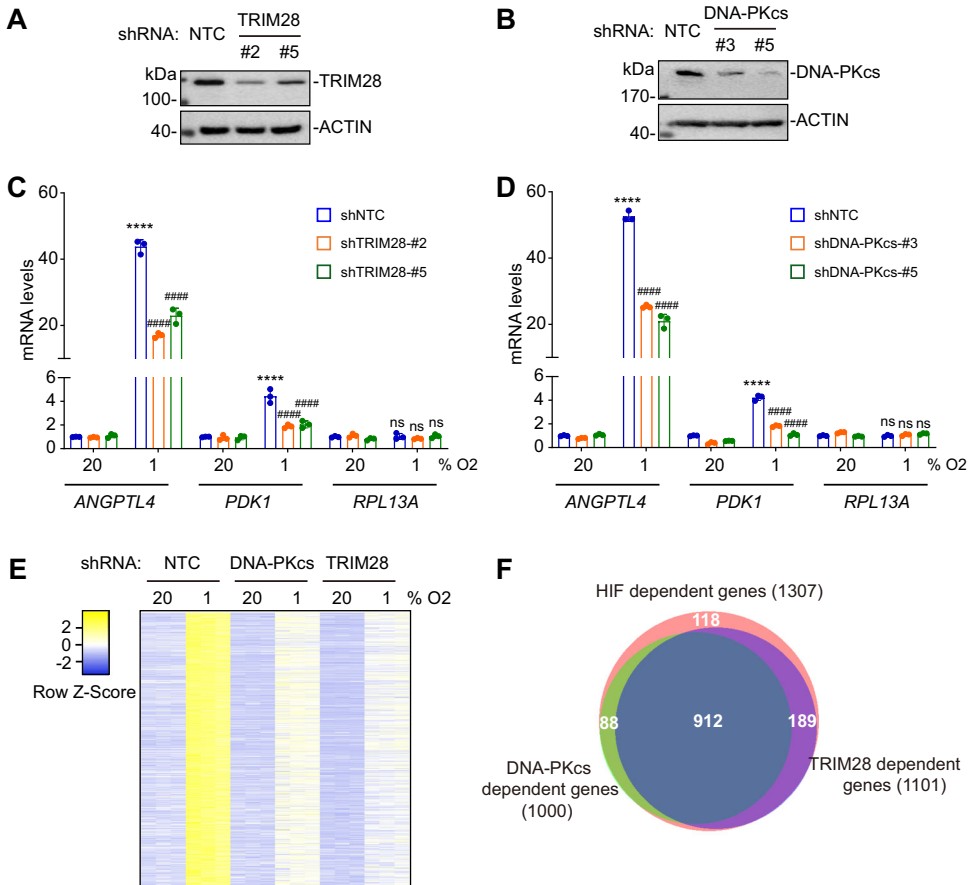

**Fig. 2 TRIM28 and DNA-PKcs are required for HIF target gene expression. A**, **B** SUM159 cells were stably transfected with NTC vector or vector encoding either of two different TRIM28 (**A**) or DNA-PKcs (**B**) shRNAs and immunoblot assays were performed. **C**, **D** NTC and TRIM28-KD (**C**) or DNA-PKcs-KD (**D**) subclones were exposed to 20% or 1% O$_2$ for 24 h and RT-qPCR assays were performed to analyze expression of HIF target (*ANGPTL4, PDK1*) and non-HIF-target (*RPL13A*) genes. For each primer pair, the qPCR data were normalized to the mean result for the NTC subclone at 20% O$_2$. Values were expressed as the mean ± SEM of three biologically independent samples. ****$p < 0.0001$ vs. NTC at 20% O$_2$; ####$p < 0.0001$ vs. NTC at 1% O$_2$; ns, not significant ($p > 0.05$) by two-way ANOVA Tukey's multiple comparisons. Exact $p$ values from left to right: <0.0001 (**C**, ANGPTL4 and PDK1), 0.9994, 0.625, >0.999 (**C**, RPL13A); < 0.0001 (**D**, ANGPTL4 and PDK1), >0.999, 0.569, 0.0667 (**D**, RPL13A). **E** RNA-seq analysis of HIF-dependent, hypoxia-induced mRNA expression in NTC, TRIM28-KD, DNA-PKcs-KD subclones of SUM159 exposed to 20% or 1% O$_2$ for 24 h is shown. **F** Most hypoxia-induced gene expression in SUM159 cells that was HIF-dependent was also TRIM28- and/or DNA-PKcs-dependent.

1α-dependent, we analyzed SUM159 subclones expressing NTC shRNA or shRNA targeting HIF-1α. Hypoxia-induced recruitment of TRIM28 and DNA-PKcs was significantly decreased in cells expressing shRNA targeting HIF-1α (Supplementary Fig. 2). Thus, HIFs coordinate the formation of a multi-protein complex with TRIM28 and DNA-PK in response to hypoxia.

**TRIM28 and DNA-PKcs are required for HIF target gene expression.** To investigate the role of TRIM28 and DNA-PK in HIF target gene expression, we stably transfected SUM159 cells with expression vectors encoding NTC shRNA or either of two shRNAs targeting TRIM28 (Fig. 2A) or DNA-PKcs (Fig. 2B). These knockdown (KD) subclones were then exposed to 20% or 1% O$_2$ for 24 h and expression of ANGPTL4 and PDK1 mRNA was analyzed by reverse transcription (RT) and qPCR. Hypoxia-induced expression of ANGPTL4 and PDK1 mRNA was observed in NTC cells but was significantly attenuated in TRIM28-KD (Fig. 2C) and DNA-PKcs-KD (Fig. 2D) subclones, whereas expression of a non-HIF target gene, *RPL13A*, was not affected. Similar results were obtained using TRIM28-KD and DNA-PKcs-KD subclones of MDA-MB-231 (Supplementary Fig. 3A–D).

To establish a broader role for TRIM28 and DNA-PK in HIF target gene transcription, we performed RNA sequencing (RNA-seq) in SUM159 cells. NTC, HIF-DKD, TRIM28-KD, and DNA-PKcs-KD subclones were exposed to 20% or 1% O$_2$ for 24 h and three biological replicates for each condition were subjected to RNA-seq. Hypoxia significantly increased the expression of 1307 mRNAs by at least 1.5-fold (FDR <0.05) in a HIF-dependent manner (Supplementary Data 1). Hypoxia-induced expression of 1000 (77%) of these mRNAs was also DNA-PKcs-dependent and 1,101 (84%) were TRIM28-dependent (Fig. 2E, F and Supplementary Fig. 3E). Gene ontology analysis revealed that the 912 HIF/TRIM28/DNA-PK-dependent mRNAs were enriched for mediators of glycolysis and angiogenesis (Supplementary Fig. 3E), which are critical to maintenance of oxygen homeostasis. Gene expression is also repressed by hypoxia in a HIF-dependent manner, but this effect is indirect, through the HIF-dependent expression of genes encoding transcriptional repressors and microRNAs[26,27]. BHLHE40 (also known as DEC1 or SHARP2), MXI1, TWIST1, and ZEB2 (also known as SIP1 or ZFHX1B) are transcriptional repressors that were induced by hypoxia in a HIF-dependent manner in SUM159 cells. The RNA-seq platform was not optimized for detection of microRNAs but MIR29B2CHG,

MIR31HG, MIR34AHG, MIR155HG, MIR210, MIR210HG, MIR570, and MIR3125 were induced by hypoxia in a HIF-dependent manner. Further studies are required to determine the extent to which each of these repressors and microRNAs is responsible for the observed changes in mRNA expression. Expression of 817 genes was hypoxia-repressed in a HIF-dependent manner and, of these, 618 (76%) were DNA-PKcs-dependent and 707 (87%) were TRIM28-dependent (Supplementary Fig. 3F). Taken together, these data demonstrate that both TRIM28 and DNA-PKcs are required for HIF-dependent regulation of gene expression in human breast cancer cells.

**TRIM28 and DNA-PK are required for HIF-1 transcriptional activity.** NTC, TRIM28-KD, and DNA-PKcs-KD subclones of SUM159 cells were co-transfected with HIF-dependent reporter plasmid p2.1, which contains the *ENO1* HRE and a basal SV40 promoter upstream of firefly luciferase (Fluc) coding sequences, and control reporter pSV-RL, which contains the same SV40 promoter upstream of Renilla luciferase (Rluc) coding sequences (Fig. 3A). Transfected cells were exposed to 20% or 1% $O_2$ for 24 h, and the Fluc/Rluc ratio was determined as a measure of HIF

activity. Knockdown of endogenous TRIM28 or DNA-PK expression significantly decreased HIF transcriptional activity under hypoxic conditions (Fig. 3B).

To determine whether TRIM28 and DNA-PK directly stimulate HIF-1α transactivation domain (TAD) function, SUM159 subclones were co-transfected with: pGalA, which encodes the yeast GAL4 DNA-binding domain fused to HIF-1α amino acids 531–826, which encompass the TADs[28], or pGalO, which encodes only the GAL4 DNA-binding domain; reporter plasmid pG5E1bLuc, which contains five GAL4-binding sites and the adenovirus *E1b* gene basal promoter upstream of Fluc coding sequences; and control reporter pSV-RL (Fig. 3C). Exposure of transfected NTC cells to 1% $O_2$ for 24 h increased Fluc activity mediated by GalA, but not by GalO; however, hypoxic induction of GalA-dependent transcription was significantly decreased in TRIM28-KD and DNA-PKcs-KD cells (Fig. 3D). Thus, HIF-1α residues 531–826, which encompass the TADs, physically interact with TRIM28 (Fig. 1F) and DNA-PK subunit KU86 (Supplementary Fig. 1C), and HIF-1α TAD function is dependent upon TRIM28 and DNA-PKcs (Fig. 3D).

Next, we investigated whether interaction with TRIM28 and DNA-PKcs stabilizes HIF binding at HREs. ChIP-qPCR assays

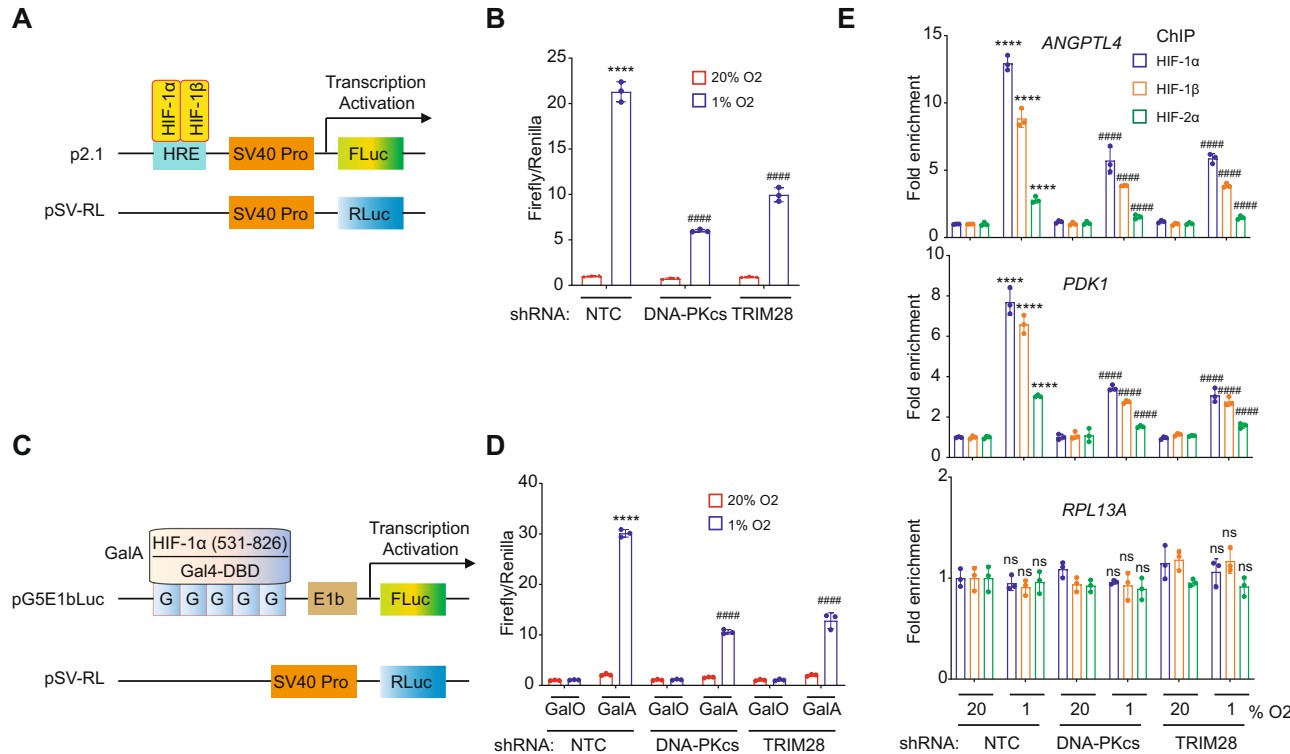

**Fig. 3 TRIM28 and DNA-PK are required for HIF-1 transcriptional activity. A, B** NTC, TRIM28 and DNA-PKcs-KD subclones of SUM159 cells were transiently co-transfected with HIF-1-dependent firefly luciferase (FLuc) reporter plasmid p2.1 and control Renilla luciferase (RLuc) reporter plasmid pSV-RL (**A**). At 24 h post-transfection, the cells were exposed to 20% or 1% $O_2$ for 24 h, lysed, and the FLuc/RLuc ratio was determined (**B**). Values were expressed as the mean ± SEM of three biologically independent samples. ****$p < 0.0001$ vs. NTC at 20% $O_2$; ####$p < 0.0001$ vs. NTC at 1% $O_2$ by two-way ANOVA Tukey's multiple comparisons. Exact $p$ values from left to right: <0.0001, <0.0001, <0.0001. **C, D** SUM159 subclones were transiently co-transfected with expression vector pGalO (encoding GAL4 DNA-binding domain) or pGalA (encoding GAL4 DNA-binding domain fused to amino acid residues 531–826 of HIF-1α), FLuc reporter pG5E1bLuc containing five GAL4 binding sites, and pSV-RL (**C**). At 24 h post-transfection, the cells were exposed to 20% or 1% $O_2$ for 24 h and FLuc/RLuc ratio was determined (**D**). Values were expressed as the mean ± SEM of three biologically independent samples. ****$p < 0.0001$ vs. NTC cells transfected with pGalO at 20% $O_2$; ####$p < 0.0001$ vs. NTC cells transfected with pGalO at 1% $O_2$ by two-way ANOVA Tukey's multiple comparisons. Exact $p$ values from left to right: <0.0001, <0.0001, <0.0001. **E** SUM159 subclones were exposed to 20% or 1% $O_2$ for 16 h and ChIP assays were performed using HIF-1α, HIF-1β, or HIF-2α Ab. Primers encompassing the *ANGPTL4* or *PDK1* HRE were used for qPCR, and *RPL13A* was analyzed as a negative control. For each primer pair, qPCR data were normalized to the mean value for NTC cells exposed to 20% $O_2$. Values were expressed as the mean ± SEM of three biologically independent samples. ****$p < 0.0001$ vs. NTC at 20% $O_2$; ####$p < 0.0001$ vs. NTC at 1% $O_2$; ns, not significant ($p > 0.05$) by two-way ANOVA Tukey's multiple comparisons. Exact $p$ values from left to right: <0.0001 (ANGPTL4 and PDK1); 0.9905, 0.8949, 0.9957, >0.999, >0.999, 0.9391, 0.7733, 0.0892, 0.9906 (RPL13A).

revealed a significant decrease in the hypoxia-induced occupancy by HIF-1α, HIF-1β, and HIF-2α of the *ANGPTL4* and *PDK1* HREs (but not the *RPL13A* gene) in TRIM28-KD and DNA-PKcs-KD subclones of SUM159 (Fig. 3E) and MDA-MB-231 (Supplementary Fig. 4A). Importantly, neither TRIM28-KD nor DNA-PKcs-KD altered HIF-1α, HIF-1β, or HIF-2α protein levels (Supplementary Fig. 4B). These data suggest that physical interactions with TRIM28 and DNA-PK stabilize HIF occupancy of HREs, thereby contributing to HIF transcriptional activity in hypoxic breast cancer cells.

**DNA-PK catalytic activity is required for HIF target gene expression.** The kinase activity of DNA-PK is critical for its function in the repair of DNA double-strand breaks (DSBs) by non-homologous end joining[29]. Formation of the DNA-PK holoenzyme requires the interaction of DNA-PKcs with the regulatory subunits, KU70 and KU86[30]. Hypoxia was reported to induce autophosphorylation of DNA-PKcs in the absence of DSBs[31]. To determine whether the catalytic activity of DNA-PK is required for regulation of HIF transactivation, we performed co-IP assays, which confirmed that KU70 and KU86 interact constitutively, whereas their interaction with DNA-PKcs was induced by hypoxia (Fig. 4A, B). Furthermore, we observed significantly impaired hypoxic induction of *ANGPTL4* and *PDK1* gene expression in KU70-KD (Fig. 4C, D) and KU86-KD (Fig. 4E, F) SUM159 subclones. Similar results were observed in KU70-KD (Supplementary Fig. 5A, B) and KU86-KD (Supplementary Fig. 5C, D) subclones of MDA-MB-231. DNA-PKcs co-immunoprecipitated with KU86 from hypoxic NTC cells but the co-IP was less efficient in TRIM28-KD cells (Supplementary Fig. 5E). These data indicate that hypoxia induces DNA-PK heterotrimer formation, and that KU70 and KU86, which are required for the catalytic activity of DNA-PK, are also required for HIF target gene expression.

**DNA damage and ATM activation are not induced by exposure of breast cancer cells to 1% O₂.** Histone H2AX is rapidly phosphorylated at S139 (γH2AX) by ATM, which is activated by autophosphorylation at S1981 in response to DNA damage[32,33]. Given that DNA-PK plays a vital role in DSB repair and DSBs have been implicated in the heat shock transcriptional response[19], we tested whether hypoxia causes DNA damage in breast cancer cells. Immunoblot assays of MDA-MB-231 cells exposed to 1% O₂ for 0, 8, 16, or 24 h, or treated with the clastogen doxorubicin at 0.5 μM for 24 h as a positive control for DNA damage, revealed that two markers of DNA DSBs, γH2AX and phosphorylation of ATM at S1981, were detected in cells exposed to doxorubicin, but not in hypoxic cells (Fig. 4G). To further investigate whether hypoxia induced a DNA damage response, we analyzed SUM159 cells exposed to 1% O₂ for 0, 15, 30, or 60 min. Hypoxia induced phosphorylation of TRIM28 at S824 and DNA-PKcs at T2609 within 15 min, in parallel with HIF-1α protein accumulation (Supplementary Fig. 6A). However, there was no evidence of phosphorylation of H2AX at S139 (γH2AX) or ATM at S1981, which suggested that 1% O₂ did not induce a DNA damage response in these cells. We also investigated whether hypoxia induced γH2AX specifically at HREs. ChIP-qPCR assays using an antibody against γH2AX did not reveal increased binding at either the *ANGPTL4* or *PDK1* HRE under hypoxic conditions (Supplementary Fig. 6B).

In agreement with these data, KU55933, an ATM inhibitor, had no effect on hypoxia-induced HIF target gene expression (Supplementary Fig. 6C). Hypoxia-induced phosphorylation of DNA-PKcs at T2609 and TRIM28 on S824 was blocked by the DNA-PK inhibitor NU7441 but not by the ATM inhibitor KU55933 (Fig. 5A). Finally, hypoxia-induced occupancy of HREs

by phospho-S824-TRIM28 and phospho-T2609-DNA-PKcs was blocked by NU7441 but not by KU55933 (Fig. 5B). These results indicate that exposure of breast cancer cells to 1% O₂ does not cause detectable DNA damage or increased ATM activity. In contrast, ATM is required for heat shock-induced gene expression and has been implicated in HIF-1 dependent hypoxia signaling[18,19,34,35], although it appears that DNA damage occurs only at O₂ levels below 0.1%[36–38].

In the DSB pathway, activation of the DNA-PK heterotrimer leads to autophosphorylation of DNA-PKcs at S2056 and threonine-2609 (T2609), which leads to conformational changes of DNA-PKcs that are required for interaction with DNA and other DNA repair proteins[39,40]. In contrast, hypoxia induced the phosphorylation of DNA-PKcs at T2609 but not at S2056 (Fig. 4G). Hypoxia-induced phosphorylation at T2609 was inhibited by the DNA-PKcs inhibitor NU7441 (Fig. 4H), indicating autophosphorylation. Hypoxia-induced expression of the HIF target genes *ANGPTL4* and *PDK1*, but not *RPL13A*, was also inhibited by NU7441 at two doses (Fig. 4I). Finally, compared to NTC cells, hypoxia-induced recruitment of all three DNA-PK subunits to HREs (Fig. 4J) and DNA-PKcs autophosphorylation (Fig. 4K) were inhibited in TRIM28-KD cells. Taken together, these results demonstrate that hypoxia induces TRIM28- and KU70/86-dependent, but ATM- and DNA damage-independent, activation of DNA-PKcs, which is required for HIF target gene expression in human breast cancer cells.

**TRIM28 phosphorylation by DNA-PK is critical for HIF target gene expression.** Given that the catalytic activity of DNA-PK is important for HIF target gene expression (Fig. 4I) and TRIM28 phosphorylation at S824 by DNA-PK is critical for the heat shock transcriptional response[18,19], we next investigated whether TRIM28 is phosphorylated by DNA-PK in response to hypoxia. SUM159 cells were exposed to 20% or 1% O₂ for 8 h and immunoblot assays revealed that hypoxia induced the phosphorylation of DNA-PKcs at T2609 and TRIM28 at S824 with similar kinetics (Supplementary Fig. 6A) and in a HIF-dependent manner (Fig. 5C). Hypoxia-induced phosphorylation of TRIM28 at S824 was eliminated by knockdown of KU70, KU86 or DNA-PKcs (Fig. 5D), or by treatment with the DNA-PK inhibitor NU7441 (Fig. 5A).

ChIP-qPCR assays revealed similarly robust hypoxia-induced occupancy of the *ANGPTL4* and *PDK1* HREs, but not the *RPL13A* gene, using antibody against either total TRIM28 or phospho-S824-TRIM28 in both SUM159 (Fig. 5E) and MDA-MB-231 (Supplementary Fig. 7A) cells. To further investigate the importance of the phosphorylation of TRIM28 at S824, we engineered two TRIM28 mutants: TRIM28-S824A, which cannot be phosphorylated, and TRIM28-S824D, in which introduction of the negatively charged aspartate residue mimics constitutive phosphorylation. We stably transfected a TRIM28-KD subclone with empty vector (EV) or vector encoding TRIM28-WT, TRIM28-S824A or TRIM28-S824D, which were expressed at similar levels (Fig. 5F). RT-qPCR assays revealed that TRIM28-WT rescued hypoxia-inducible *ANGPTL4* and *PDK1* gene expression, whereas TRIM28-S824A did not (Fig. 5G). Remarkably, HIF target gene expression was increased at both 20% and 1% O₂ in the TRIM28-S824D subclone (Fig. 5G), demonstrating that introduction of a negative charge at residue 824 of TRIM28 (by DNA-PK-dependent serine phosphorylation or aspartate substitution) is critical for activation of HIF target gene expression. As predicted by our model, the DNA-PK inhibitor NU7441 blocked HIF target gene activation in WT-TRIM28-expressing cells but not in TRIM28-S824D-expressing cells (Supplementary Fig. 7B).

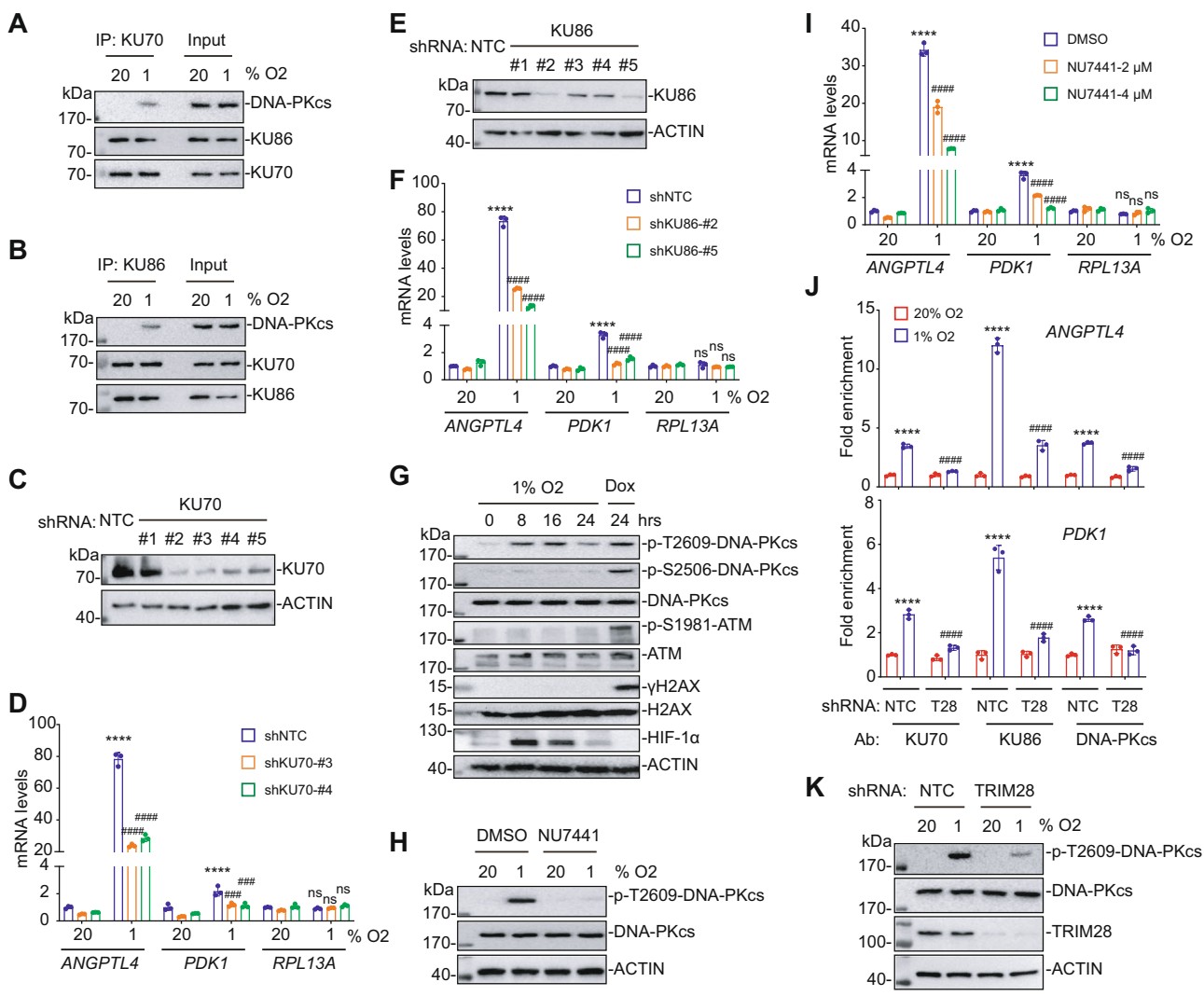

**Fig. 4 DNA-PK heterotrimer activity is required for HIF target gene expression. A**, **B** SUM159 cells were exposed to 20% or 1% $O_2$ for 8 h and IP was performed with KU70 (**A**) or KU86 (**B**) Ab, followed by immunoblot assays. **C** SUM159 cells were stably transfected with NTC or a KU70 shRNA vector and immunoblot assays were performed. **D** NTC and KU70-KD subclones were exposed to 20% or 1% $O_2$ for 24 h and RT-qPCR assays were performed. For each primer pair, the qPCR data were normalized to the mean result for NTC cells at 20% $O_2$. Values were expressed as the mean ± SEM of three biologically independent samples. ****$p < 0.0001$ vs. NTC at 20% $O_2$; ###$p < 0.001$, ####$p < 0.0001$ vs. NTC at 1% $O_2$; ns, not significant ($p > 0.05$) by two-way ANOVA Tukey's multiple comparisons. Exact $p$ values from left to right: <0.0001 (ANGPTL4); <0.0001, 0.0004, 0.0002 (PDK1); 0.7126, 0.9872, 0.0601 (RPL13A). **E** SUM159 cells were stably transfected with NTC or a KU86 shRNA vector and immunoblot assays were performed. **F** NTC and KU86-KD subclones were exposed to 20% or 1% $O_2$ for 24 h and RT-qPCR assays were performed. Values were expressed as the mean ± SEM of three biologically independent samples. ****$p < 0.0001$ vs. NTC at 20% $O_2$; ####$p < 0.0001$ vs. NTC at 1% $O_2$; ns, not significant ($p > 0.05$) by two-way ANOVA Tukey's multiple comparisons. Exact $p$ values from left to right: <0.0001 (ANGPTL4 and PDK1); 0.8327, 0.4358, 0.5145 (RPL13A). **G** SUM159 cells were exposed to 1% $O_2$ for 0–24 h or 0.5 μM doxorubicin for 24 h (Dox) and immunoblot assays were performed. **H** SUM159 cells were exposed to 20% or 1% $O_2$ for 8 h in the presence of DMSO or NU7441 and immunoblot assays were performed. **I** SUM159 cells were exposed to 20% or 1% $O_2$ for 24 h in the presence of DMSO or NU7441 and RT-qPCR assays were performed. Values were expressed as the mean ± SEM of three biologically independent samples. ****$p < 0.0001$ vs. NTC at 20% $O_2$; ####$p < 0.0001$ vs. NTC at 1% $O_2$; ns, not significant ($p > 0.05$) by two-way ANOVA Tukey's multiple comparisons. Exact $p$ values from left to right: <0.0001 (ANGPTL4 and PDK1); 0.3696, 0.9846, 0.2396 (RPL13A). **J** NTC and TRIM28-KD (T28) subclones were analyzed by ChIP-qPCR with the indicated Ab. For each primer pair, the qPCR data were normalized to the mean result for NTC cells at 20% $O_2$. Values were expressed as the mean ± SEM of three biologically independent samples. ****$p < 0.0001$ vs. NTC at 20% O2; ####$p < 0.0001$ vs. NTC at 1% O2 by two-way ANOVA Tukey's multiple comparisons. Exact $p$ values: <0.0001 (ANGPTL4 and PDK1). **K** NTC and TRIM28-KD subclones were exposed to 20% or 1% $O_2$ for 8 h and immunoblot assays were performed.

TRIM28 was shown to function as an E3 SUMO ligase and two critical cysteine residues in its RING domain (C65 and C68) were critical for its catalytic activity[41]. To determine whether E3 SUMO ligase activity is required for TRIM28-mediated HIF transactivation, we stably expressed TRIM28-C65A/C68A or WT-TRIM28 in TRIM28-KD MDA-MB-231 cells and demonstrated similar protein levels by immunoblot assay (Supplementary Fig. 7C). Similar to TRIM28-WT, TRIM28-C65A/C68A restored hypoxia-induced HIF target gene expression (Supplementary Fig. 7D), demonstrating that E3 SUMO ligase activity is not required for TRIM28 to support HIF transactivation.

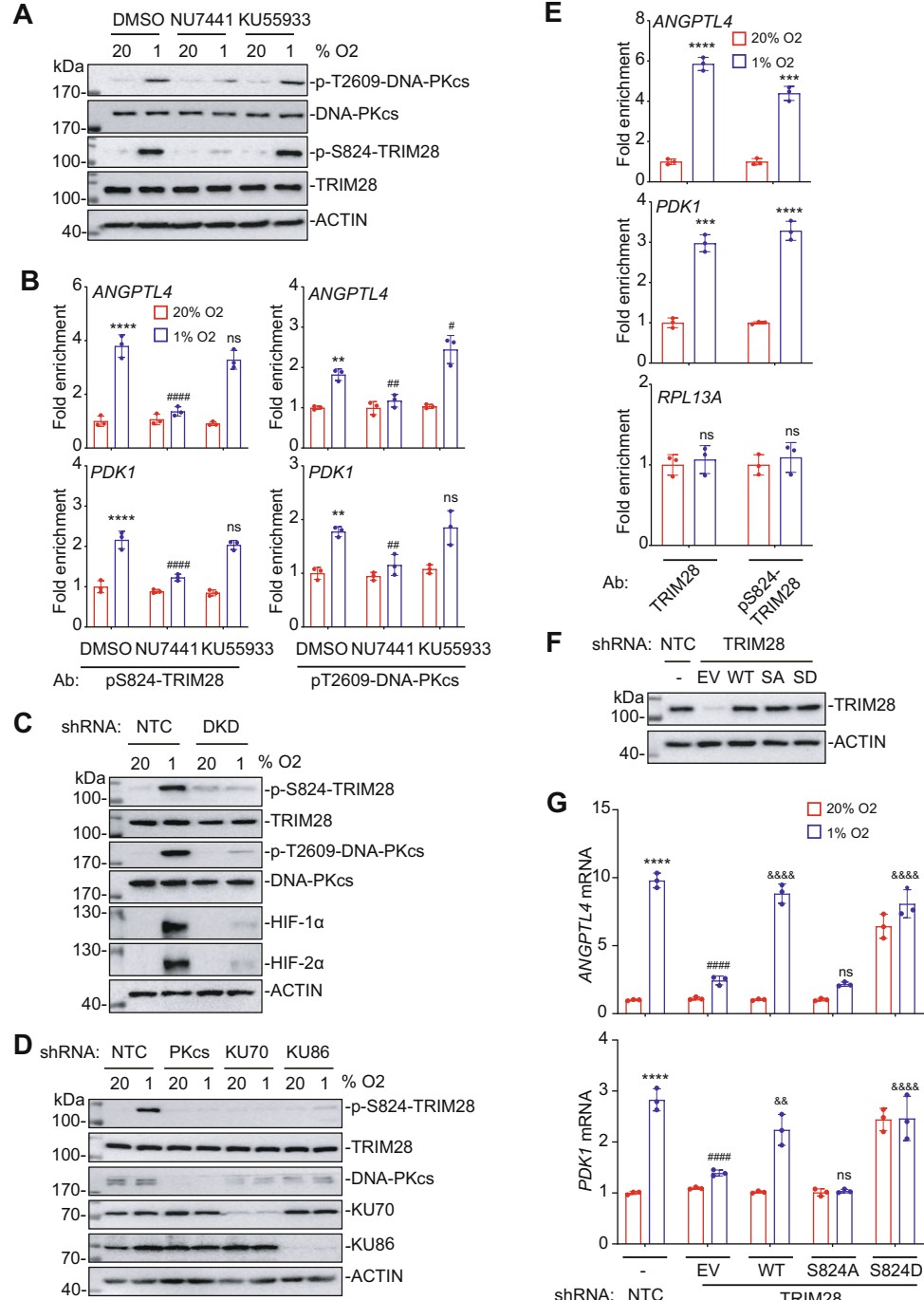

**TRIM28 and DNA-PK are required for release of paused Pol II in response to hypoxia**. Promoter-proximal pausing of Pol II is a major checkpoint in transcription of HIF target genes[13,26]. Pol II that has initiated transcription and paused is characterized by S5 phosphorylation of the CTD (S5P), whereas S2 phosphorylation (S2P) is associated with Pol II pause release[16]. Several studies have demonstrated hypoxia-induced RNA Pol II CTD phosphorylation or CDK9 recruitment to HREs[11,13,42,43]. To determine whether DNA-PKcs or TRIM28 regulates release of paused Pol II during HIF-mediated transactivation, we performed ChIP-qPCR assays after exposure of cells to 20% or 1% $O_2$ for 16 h. Hypoxia significantly increased occupancy of the *ANGPTL4* and *PDK1* gene HREs in NTC cells by total Pol II, Pol II-S2P and Pol II-S5P, but only Pol II-S2P occupancy was significantly decreased

in TRIM28-KD or DNA-PKcs-KD subclones of SUM159 (Fig. 6A) and MDA-MB-231 (Supplementary Figure 8A), indicating that TRIM28 and DNA-PK are required for phosphorylation of paused Pol II at S2 in response to hypoxia. The loss of hypoxia-induced HRE occupancy by Pol II-S2P, but not by total Pol II or Pol II-S5P in the TRIM28-KD and DNA-PKcs-KD subclones (Fig. 6A) indicates that TRIM28 and DNA-PK promote the release of paused Pol II.

NELF, which is a complex containing five subunits, binds to Pol II to induce pausing, and Pol II release and subsequent productive transcriptional elongation require the phosphorylation and dissociation of NELF[16,17]. ChIP-qPCR using an antibody against the NELF-E subunit revealed a hypoxia-induced decrease in HRE occupancy in the NTC subclone, but not in the TRIM28-

**Fig. 5 TRIM28 phosphorylation by DNA-PK is required for HIF target gene expression. A** SUM159 cells were exposed to 20% or 1% $O_2$ for 8 h, in the presence of vehicle (DMSO), NU7441 (4 μM) or KU55933 (20 μM) and aliquots of cell lysates were subjected to immunoblot assays. **B** SUM159 cells were exposed to 20% or 1% $O_2$ for 16 h, in the presence of vehicle (DMSO) or NU7441 (4 μM) or KU55933 (20 μM). ChIP assays were performed using Ab against phospho-S824-TRIM28 (left panels) or phospho-T2609-DNA-PKcs (right panels) and qPCR was performed using primers specific for the *ANGPTL4* (upper panels) or *PDK1* (lower panels) HRE. The qPCR data were normalized to the result obtained for DMSO-treated cells at 20% $O_2$. Values were expressed as the mean ± SEM of three biologically independent samples. ** $p < 0.01$, **** $p < 0.0001$ vs. NTC at 20% $O_2$; # $p < 0.05$, ## $p < 0.01$, #### $p < 0.0001$ vs. NTC at 1% $O_2$; ns, not significant ($p > 0.05$) by two-way ANOVA Tukey's multiple comparisons. Exact *p* values from left to right: <0.0001, <0.0001, 0.2326, 0.0013, 0.0091, 0.0117 (ANGPTL4); <0.0001, <0.0001, 0.8254, 0.0012, 0.0073, 0.993 (PDK1). **C, D** SUM159 subclones expressing the indicated shRNA were exposed to 20% or 1% $O_2$ for 8 h, followed by immunoblot assays. **E** SUM159 cells were exposed to 20% or 1% $O_2$ for 16 h and ChIP was performed using the indicated Ab. qPCR assays were performed using *ANGPTL4*, *PDK1*, or *RPL13A* primers with results normalized to NTC at 20% $O_2$. Values were expressed as the mean ± SEM of three biologically independent samples. *** $p < 0.001$, **** $p < 0.0001$; ns, not significant ($p > 0.05$) by two-tailed Student's *t*-test. Exact *p* values from left to right: <0.0001, 0.0001 (ANGPTL4); 0.0002, <0.0001 (PDK1); 0.624, 0.516 (RPL13A). **F, G** TRIM28-KD cells were stably transfected with empty vector (EV) or vector encoding TRIM28-WT, TRIM28-S824A or TRIM28-S824D. Immunoblot (**F**) and RT-qPCR (**G**) assays were performed. For each primer pair, the qPCR data were normalized to the mean result for NTC cells at 20% $O_2$. Values were expressed as the mean ± SEM of three biologically independent samples. **** $p < 0.0001$ vs. NTC at 20% $O_2$; #### $p < 0.0001$ vs. NTC at 1% $O_2$; && $p < 0.01$, &&&& $p < 0.0001$ vs. shTRIM28-EV at 1% $O_2$; ns, not significant ($p > 0.05$) by two-way ANOVA Tukey's multiple comparisons. Exact *p* values from left to right: <0.0001, <0.0001, <0.0001, 0.9996, <0.0001 (ANGPTL4); <0.0001, <0.0001, 0.0012, 0.4782, <0.0001 (PDK1).

KD or DNA-PKcs-KD subclone of SUM159 (Fig. 6B) and MDA-MB-231 (Supplementary Fig. 8B) cells, indicating that TRIM28 and DNA-PKcs are required for NELF expulsion from HIF target genes in response to hypoxia.

To directly analyze HIF target gene transcript elongation, we performed RT-qPCR assays of NTC cells using primers designed to amplify sequences at the 5′ end (within 60 nucleotides of the transcription start site), middle and 3′ end of ANGPTL4 mRNA. For each primer pair, the qPCR data are normalized to the mean value for the NTC subclone at 20% $O_2$. The experiment was designed to analyze the effect of hypoxia and DNA-PK or TRIM28 knockdown on transcript abundance. The experimental design does not allow comparison of the relative levels of the three different qPCR products because of potential differences in primer efficiency. Hypoxia induced only a modest increase in the abundance of transcripts containing sequences at the 5′ end of the mRNA (Fig. 6C), which supports the model that transcription initiation occurs regardless of $O_2$ concentration[13,27]. In contrast, the abundance of transcripts containing sequences in the middle and at the 3′ end of the mRNA was markedly increased when NTC cells were subjected to hypoxia (Fig. 6C). Knockdown of TRIM28 or DNA-PKcs in SUM159 cells did not affect abundance of transcripts containing sequences at the 5′ end of the mRNA but significantly decreased the abundance of transcripts containing sequences in the middle and at the 3′ end of ANGPTL4 mRNA specifically under hypoxic conditions (Fig. 6C). Similar results were obtained using MDA-MB-231 subclones (Supplementary Fig. 8C). These data provide further evidence that TRIM28 and DNA-PKcs are required for Pol II release and productive transcriptional elongation of HIF target genes in response to hypoxia.

Histone modifications play an important role in transcriptional regulation. The H3K4me3 chromatin mark is associated with transcription initiation, whereas H3K36me3 marks transcriptional elongation[44,45]. ChIP-qPCR assays using antibodies against H3K4me3, H3K36me3, and total H3 revealed that exposure of SUM159 cells to hypoxia induced a modest increase in H3K4me3 marks and a more dramatic increase in H3K36me3 marks at the *PDK1* HRE in NTC cells (Fig. 6D). Knockdown of TRIM28 or DNA-PKcs had no effect on H3K4me3 marks, but significantly decreased the hypoxic induction of H3K36me3 marks (Fig. 6D). None of these changes were observed at the *RPL13A* gene. Similar data were obtained using MDA-MB-231 cells (Supplementary Fig. 8D). These results are strikingly similar to the analysis of Pol II phosphorylation (Fig. 6A) and mRNA abundance (Fig. 6C) described above. Taken together, these data provide compelling

evidence that HIF-dependent recruitment of TRIM28 and DNA-PKcs to HREs is critical for release of Pol II from its paused state in response to hypoxia.

**Phosphorylation of TRIM28 by DNA-PK is required for CDK9 recruitment.** P-TEFb is required for the release of paused Pol II, which is mediated by phosphorylation of Pol II CTD-S2 and NELF by the P-TEFb catalytic subunit, CDK9[46]. To test whether phosphorylation of TRIM28 is required for CDK9 recruitment to HREs of HIF target genes, we treated SUM159 cells with DNA-PKcs inhibitor NU7441, which blocks phosphorylation of TRIM28 (Fig. 5C). CDK9 was recruited to HREs of HIF target genes (but not *RPL13A*) in response to hypoxia and NU7441 blocked CDK9 recruitment in a dose dependent manner (Fig. 7A). Co-IP assays revealed that, compared to its interaction with WT-TRIM28, CDK9 interacts weakly with TRIM28-S824A and strongly with TRIM28-S824D (Fig. 7B). In agreement with these interaction data, the hypoxia-induced recruitment of CDK9 to HREs was deficient in TRIM28-S824A reconstituted TRIM28-KD cells, whereas CDK9 was constitutively recruited to cells expressing TRIM28-S824D (Fig. 7C), demonstrating that introduction of a negative charge at residue 824 (by phosphorylation or amino acid substitution) is critical for CDK9 recruitment, just as it was critical for HIF target gene expression (Fig. 5G). Most importantly, hypoxia induced the interaction of endogenous CDK9 with endogenous TRIM28 phosphorylated at S824 (Fig. 7D).

**TRIM28 and DNA-PK are required for HIF-dependent induction of glycolytic metabolism.** *PDK1* encodes pyruvate dehydrogenase kinase 1, which plays a key role in the regulation of glucose metabolism by phosphorylating and inactivating the catalytic subunit of pyruvate dehydrogenase (PDH), the enzyme that converts pyruvate to acetyl-CoA for entry into the tricarboxylic acid cycle. Thus, under hypoxic conditions, HIF-dependent expression of PDK1 shunts substrate away from the mitochondria, thereby decreasing $O_2$ consumption[25,47]. The RNA-seq data from SUM159 cells indicated that, in addition to *PDK1*, TRIM28 and DNA-PKcs are required for hypoxia-induced and HIF-dependent expression of the *SLC2A1* and *SLC2A3* genes, which encode glucose transporters 1 and 3, as well as the *ALDOC* and *HK2* genes, which encode the glycolytic enzymes aldolase C and hexokinase 2 (Supplementary Data 1). We hypothesized that increased expression of glucose transporters, glycolytic enzymes and PDK1 should lead to increased lactate production in NTC

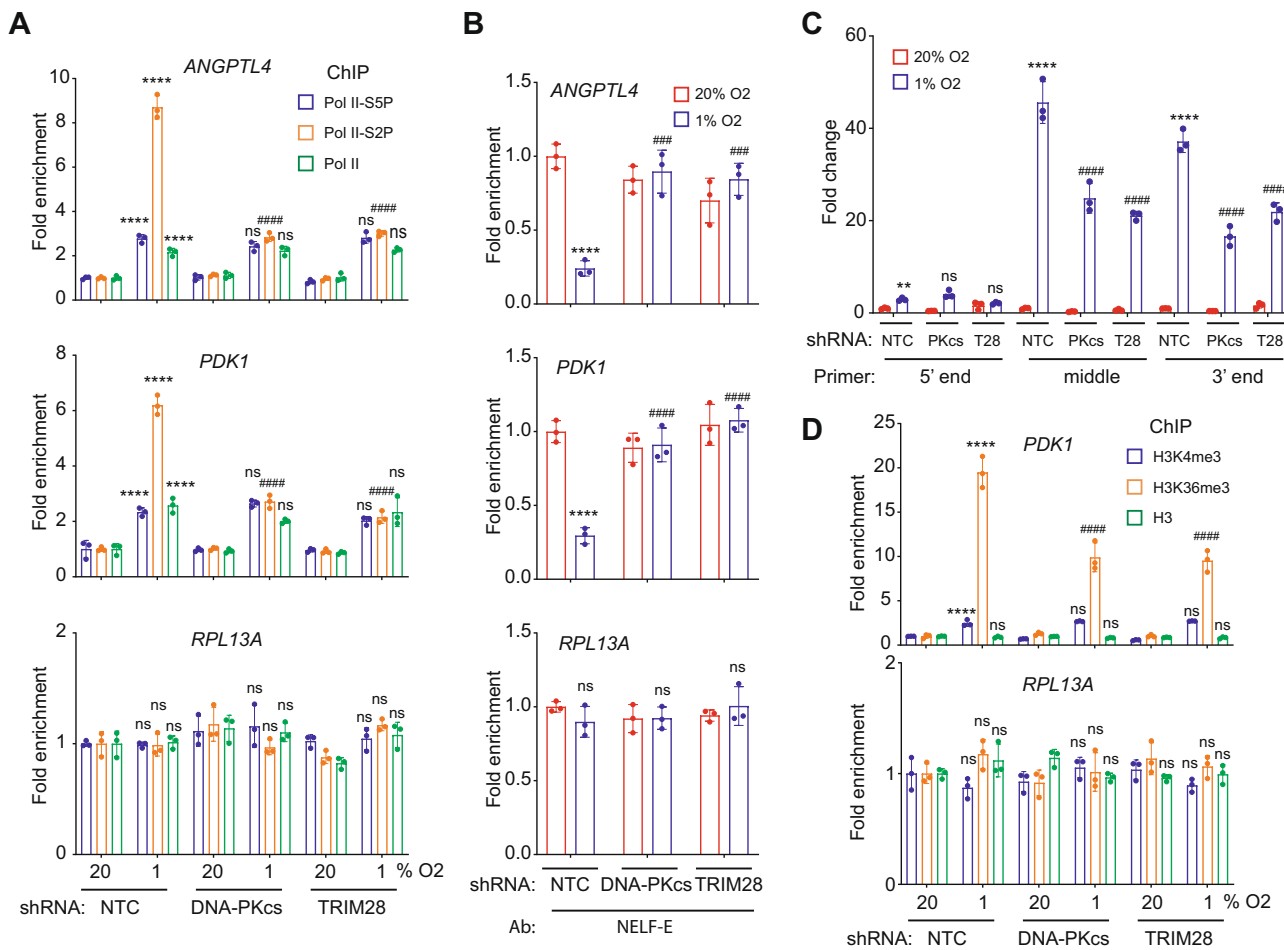

**Fig. 6 TRIM28 and DNA-PK are required for hypoxia-induced release of paused Pol II. A** SUM159 subclones were exposed to 20% or 1% $O_2$ for 16 h, ChIP was performed using Ab against Pol II-S5P, Pol II-S2P, or total Pol II, and qPCR assays were performed. For each primer pair, the qPCR data were normalized to the mean result for NTC cells at 20% $O_2$. Values were expressed as the mean ± SEM of three biologically independent samples. ****$p < 0.0001$ vs. NTC at 20% $O_2$; ####$p < 0.0001$ vs. NTC at 1% $O_2$; ns, not significant ($p > 0.05$) by two-way ANOVA Tukey's multiple comparisons. Exact $p$ values from left to right: <0.0001, <0.0001, <0.0001, 0.183, <0.0001, 0.9922, 0.9998, <0.0001, 0.9474 (ANGPTL4); <0.0001, <0.0001, <0.0001, 0.2796, <0.0001, 0.1393, 0.2735, <0.0001, 0.8443 (PDK1); >0.999, >0.999, >0.999, 0.4341, 0.999, 0.847, 0.9856, 0.2637, 0.9494 (RPL13A). **B** SUM159 subclones were analyzed by ChIP using NELF-E Ab and qPCR assays were performed. For each primer pair, the qPCR data were normalized to the mean result for NTC cells at 20% $O_2$. Values were expressed as the mean ± SEM of three biologically independent samples. ****$p < 0.0001$ vs. NTC at 20% $O_2$; ###$p < 0.001$, ####$p < 0.0001$ vs. NTC at 1% $O_2$; ns, not significant ($p > 0.05$) by two-way ANOVA Tukey's multiple comparisons. Exact $p$ values from left to right: <0.0001, 0.0001, 0.0003 (ANGPTL4); <0.0001, <0.0001, <0.0001 (PDK1); 0.7128, 0.9991, 0.6667 (RPL13A). **C** NTC, DNA-PKcs-KD (PKcs) and TRIM28-KD (T28) subclones were exposed to 20% or 1% $O_2$ for 24 h, followed by RT-qPCR assays using primers that targeted the 5' end, middle, or 3' end of ANGPTL4 mRNA. For each primer pair, the qPCR data are normalized to the mean value for the NTC subclone at 20% $O_2$. Values were expressed as the mean ± SEM of three biologically independent samples. **$p < 0.01$, ****$p < 0.0001$ vs. NTC at 20% $O_2$; ####$p < 0.0001$ vs. NTC at 1% $O_2$; ns, not significant ($p > 0.05$) by two-way ANOVA Tukey's multiple comparisons. Exact $p$ values from left to right: 0.0065, 0.1638, 0.4338 (5' end), <0.0001 (middle and 3' end). **D** Subclones were exposed to 20% or 1% $O_2$ for 16 h and ChIP-qPCR was performed using Ab against H3K4me3, H3K36me3 or total histone H3. For each primer pair, the qPCR data were normalized to the mean result for NTC cells at 20% $O_2$. Values were expressed as the mean ± SEM of three biologically independent samples. ****$p < 0.0001$ vs. NTC at 20% $O_2$; ####$p < 0.0001$ vs. NTC at 1% $O_2$; ns, not significant ($p > 0.05$) by two-way ANOVA Tukey's multiple comparisons. Exact $p$ values from left to right: <0.0001, <0.0001, 0.5509, 0.5443, <0.0001, 0.5673, 0.4177, <0.0001, 0.8219 (PDK1); 0.6161, 0.5697, 0.4981, 0.2732, 0.6499, 0.2597, 0.9997, 0.8869, 0.4512 (RPL13A).

cells under hypoxic conditions, which was confirmed by direct analysis of conditioned medium, whereas hypoxia-induced lactate levels were significantly decreased by knockdown of HIF-1α, DNA-PKcs or TRIM28 (Fig. 8A). We also hypothesized that decreased PDK1 expression in the knockdown subclones should lead to an increased $O_2$ consumption rate (OCR). Compared to NTC cells, the knockdown subclones did indeed manifest significantly increased OCR (Fig. 8B), both in terms of basal respiration (time points before oligomycin addition) and maximal respiratory capacity (time points after FCCP addition and before addition of rotenone + antimycin). Finally, gene set enrichment

analysis of RNA-seq data from 290 human breast cancers revealed significantly increased expression of glycolysis-related genes in tumors with *TRIM28* expression that was greater than the median (Fig. 8C), providing a clinical correlation that complements the mechanistic data derived from the SUM159 subclones.

## Discussion

Under hypoxic conditions, $O_2$-dependent hydroxylation, ubiquitination, and degradation of HIF-1α is inhibited, leading to rapid

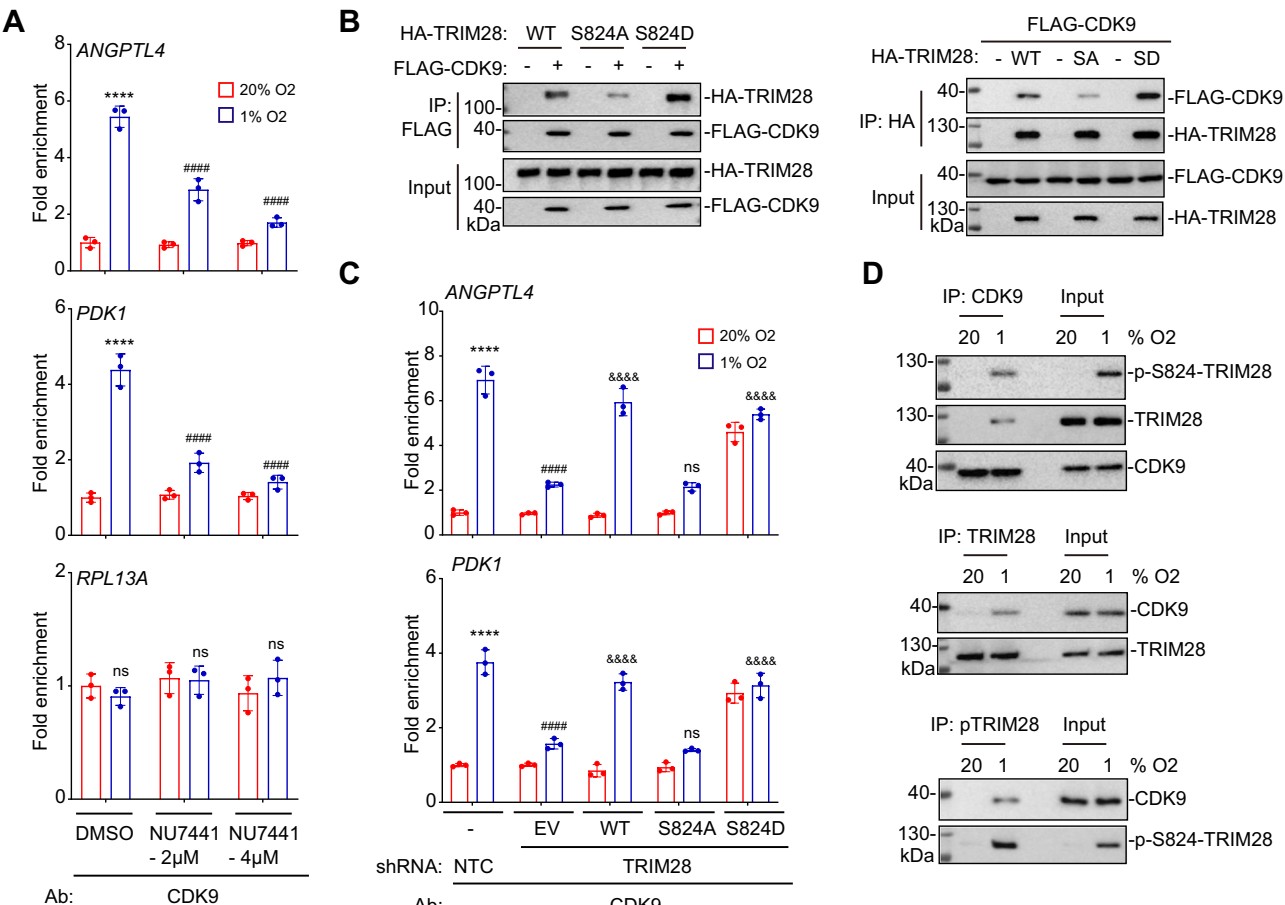

**Fig. 7 TRIM28 phosphorylation by DNA-PK induces CDK9 recruitment to HREs. A** SUM159 cells were exposed to 20% or 1% $O_2$ in the presence of DMSO or NU7441, and ChIP-qPCR assays were performed with CDK9 Ab. For each primer pair, the qPCR data were normalized to the mean result for DMSO-treated cells at 20% $O_2$. Values were expressed as the mean ± SEM of three biologically independent samples. ****$p < 0.0001$ vs. DMSO at 20% $O_2$; ####$p < 0.0001$ vs. DMSO at 1% $O_2$; ns, not significant ($p > 0.05$) by two-way ANOVA Tukey's multiple comparisons. Exact $p$ values from left to right: <0.0001 (ANGPTL4 and PDK1); 0.9447, 0.748, 0.638 (RPL13A). **B** SUM159 cells were co-transfected with expression vectors encoding FLAG-CDK9 and HA-tagged TRIM28-WT, TRIM28-S824A or TRIM28-S824D. IP with FLAG Ab (left) or HA Ab (right) was performed, followed by immunoblot assays. **C** NTC and TRIM28-KD subclones transfected with empty vector (EV) or vector encoding TRIM28-WT, TRIM28-S824A or TRIM28-S824D were exposed to 20% or 1% $O_2$ for 16 h and ChIP-qPCR assays were performed using CDK9 Ab. For each primer pair, the qPCR data were normalized to the mean result for NTC cells at 20% $O_2$. Values were expressed as the mean ± SEM of three biologically independent samples. ****$p < 0.0001$ vs. NTC at 20% $O_2$; ####$p < 0.0001$ vs. NTC at 1% $O_2$; &&&&$p < 0.0001$ vs. TRIM28-KD-EV at 1% $O_2$; ns, not significant ($p > 0.05$) by two-way ANOVA Tukey's multiple comparisons. Exact $p$ values from left to right: <0.0001, <0.0001, <0.0001, >0.999, <0.0001 (ANGPTL4); <0.0001, <0.0001, <0.0001, 0.987, <0.0001 (PDK1). **D** SUM159 cells were exposed to 20% or 1% $O_2$ for 8 h and IP was performed with CDK9 Ab (upper), total TRIM28 Ab (middle), p-S824-TRIM28 Ab (bottom), followed by immunoblot assays.

protein accumulation[2,3,6,7]. The data presented in this study demonstrate that the induction of HIF-1α protein expression in response to hypoxia leads to interaction with TRIM28. Further studies are required to determine whether TRIM28 also binds directly to DNA at HIF target genes as previously reported for the *HSPA1B* gene[18]. Together, HIF-1α and TRIM28 assemble the DNA-PK heterotrimer, which autophosphorylates DNA-PKcs on T2609 and phosphorylates TRIM28 on S824, enabling TRIM28 to recruit CDK9, which phosphorylates Pol II CTD on S2, triggering release of the paused polymerase, leading to productive transcriptional elongation of HIF target genes (Fig. 8D). We have demonstrated hypoxia-induced recruitment of TRIM28 and DNA-PK to 10 out of 10 HIF target genes analyzed by ChIP-qPCR and demonstrated by RNA-seq that the vast majority of HIF target genes require TRIM28 and DNA-PK for hypoxia-induced expression. A more comprehensive analysis will require using ChIP-seq to compare the genomic footprints of TRIM28, DNA-PK, HIF-1, and HIF-2.

The multi-domain protein TRIM28, also called KAP1 (KRAB-associated protein 1) or TIF1β (transcription intermediary factor 1β), was first reported to function as a transcriptional co-repressor through interaction with Kruppel transcription factor family (KRAB) members and may bind directly to certain DNA sequences[48–52]. TRIM28 acts as a co-repressor to maintain human cytomegalovirus in a latent state, whereas mTOR-dependent phosphorylation of S824 triggers lytic infection[53]. Recent work has identified TRIM28 as a critical factor that regulates Pol II pausing at signal-regulated genes and release by CDK9 recruitment[18,19].

The role of DNA-PK in binding to DSBs and initiating DNA repair by non-homologous end joining (NHEJ) is well known[54]. Immunoglobulin gene class-switching recombination, which requires DSB formation and NHEJ, activates DNA-PK[55]. Prior studies have reported that in response to DSBs, DNA-PKcs is phosphorylated at T2609 and TRIM28 is phosphorylated at S824, both of which are ATM-dependent and critical for their function

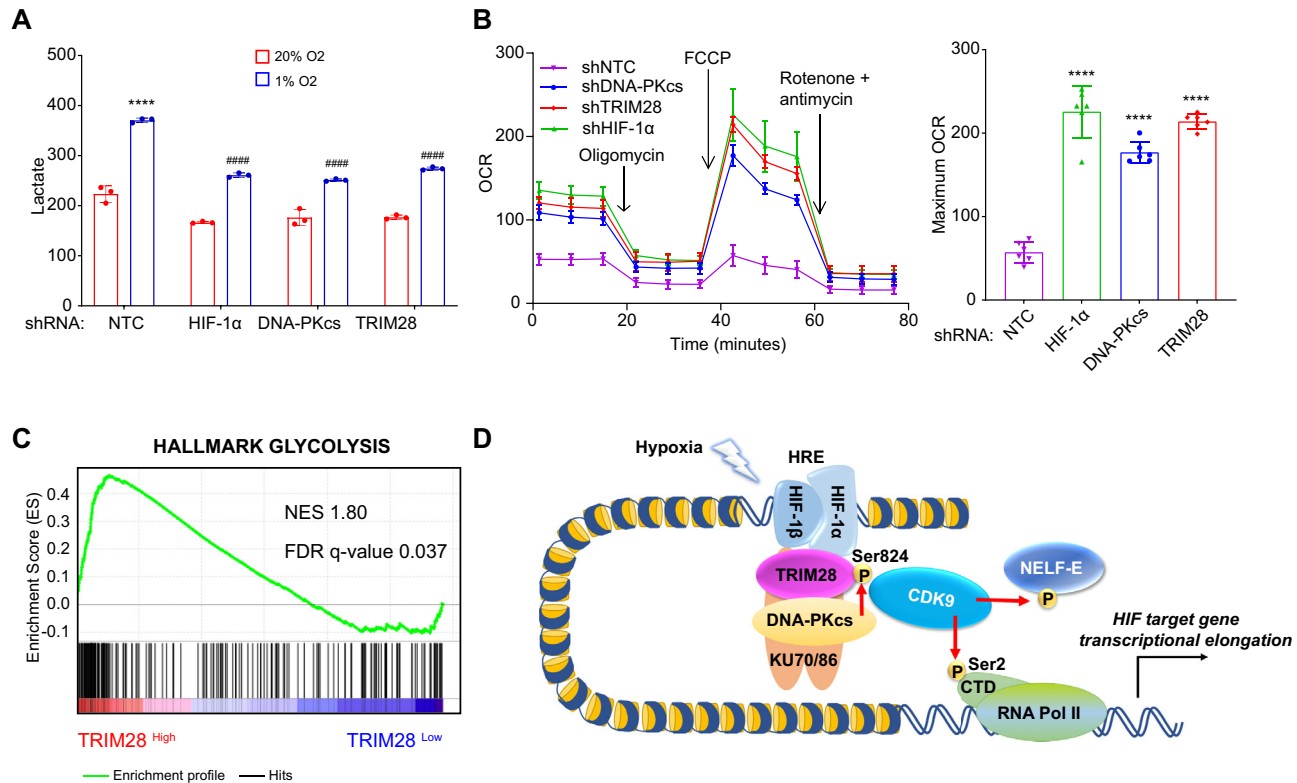

**Fig. 8 HIF-1α, TRIM28, and DNA-PK stimulate glycolysis and inhibit O₂ consumption. A** MDA-MB-231 subclones were exposed to 20% or 1% O₂ for 24 h, and lactate was measured in the culture media and normalized to cell number (nmol per $5 \times 10^5$ cells). Values were expressed as the mean ± SEM of three biologically independent samples. ****$p < 0.0001$ vs. NTC at 20% O₂; ####$p < 0.0001$ vs. NTC at 1% O₂ by two-way ANOVA Tukey's multiple comparisons. Exact $p$ values from left to right: <0.0001, <0.0001, <0.0001. **B** O₂ consumption rate (OCR) was measured and normalized to cell number (pmol/min/$10^4$ cells). Values were expressed as the mean ± SEM of 6 technical repeats. ****$p < 0.0001$ vs. NTC by two-tailed Student's $t$-test. Exact $p$ values from left to right: <0.0001, <0.0001, <0.0001. **C** Gene set enrichment analysis revealed that expression of the "hallmark glycolysis" gene set was significantly correlated with TRIM28 expression in the BRCA data set from TCGA. NES, Normalized Enrichment Score. **D** HIF-1α triggers release of paused Pol II by assembling a multi-protein complex in response to hypoxia. HIF-1α recruits TRIM28 and catalytically active DNA-PK heterotrimer, which phosphorylates TRIM28 at Ser-824, enabling recruitment of CDK9, which phosphorylates the NELF-E subunit of negative elongation factor and Pol II on Ser-2 of CTD to release Pol II for productive transcriptional elongation of HIF target genes.

in DNA damage pathways[56–59]. Thus, the interaction of HIF-1α and TRIM28 appears to substitute for DNA damage as a stimulus for DNA-PK assembly and activity in response to hypoxia. Functions of DNA-PK that are outside of classical DNA-damage response pathways have been identified. A recent study demonstrated that DNA-PK binds to the U3 small nucleolar RNA, which is required for the processing of 18 S ribosomal RNA, and this interaction triggers autophosphorylation of DNA-PKcs at T2609 but not at S2506[60]. Another study reported that DNA-PKcs was autophosphorylated at S2056, independent of DSBs, in cells exposed to hypoxia (0.1–1% O₂), and that the stability of HIF-1α was decreased in a cell line lacking DNA-PKcs expression[31]. However, phosphorylation of S2056 was not observed until 12 h of hypoxic exposure and was maximal at 24 h. By contrast, we demonstrated that hypoxia activates DNA-PK activity in a KU70/KU86-, TRIM28- and HIF-dependent manner with rapid autophosphorylation at T2609, leading to the phosphorylation of TRIM28 at S824, which was required for transcriptional elongation of HIF target genes. We demonstrated progressive induction of T2609 phosphorylation, in concert with TRIM28 phosphorylation and accumulation of HIF-1α, over the course of 1 h (Supplementary Fig. 6A), and no phosphorylation of S2056 through 24 h (Fig. 4G). In addition, HIF-1α was expressed normally in DNA-PKcs-KD cells (Supplementary Fig. 4B). Whether there are cell-type-specific differences in the mechanism and consequences of DNA-PK activation in response to hypoxia

will require further study. Similarly, it is possible that other cell types may be more dependent on other mechanisms for release of paused Pol II at HIF target genes, such as involvement of the CDK8-Mediator complex[13].

The critical involvement of TRIM28 and DNA-PK in the transcriptional response to hypoxia bears similarity to the heat shock response, but with several key differences. We have established a molecular mechanism that accounts, first, for hypoxia-induced recruitment of DNA-PK specifically to HIF target genes and its activation in the absence of DNA damage. Second, we establish that the HIF-dependent co-localization of TRIM28 and active DNA-PK heterotrimer enables phosphorylation of the former by the latter. Third, unlike heat shock[18], exposure of breast cancer cells to 1% O₂ does not induce DSBs or ATM activity. This result is consistent with prior reports involving many[31,36,61–63] but not all[64] cell types. Fourth, in the heat shock response, TRIM28 is constitutively bound to the *HSPA1B* promoter and is required both for Pol II pausing at the *HSPA1B* gene in the uninduced state and for Pol II release upon heat shock[18]; by contrast, our data suggest that at HIF target genes relief of Pol II pausing, as inferred by Pol II-S2P ChIP (Fig. 6A), and transcriptional elongation, as inferred by RT-qPCR transcript mapping (Fig. 6C) and H3K36me3 ChIP (Fig. 6D), are not significantly affected by TRIM28-KD under non-hypoxic conditions, from which we conclude that TRIM28 is an essential positive regulator in the induced state and has no major effect on HIF target gene expression in the uninduced state.

Recently, TRIM28 was shown to repress HIV-1 transcription and thereby promote latency by sumoylation of CDK9, which inhibits its catalytic activity by blocking interaction with cyclin T1[65]. By contrast, our data indicate that the E3 SUMO-ligase activity of TRIM28 is dispensable for regulation of HIF target gene transcription (Supplementary Fig. 7D). Instead, the key role of TRIM28 is to form a stable complex with HIF-1 and DNA-PK, leading to its phosphorylation at S824, which is required for CDK9 recruitment. There is evidence that phosphorylation of TRIM28 on S824 results in desumoylation[66], but the molecular mechanisms regulating the E3 SUMO-ligase activity of TRIM28 require further study.

In this paper we demonstrate that HIFs recruit TRIM28 and DNA-PK to HREs, and that TRIM28 and DNA-PK recruitment is required for stable HIF occupancy. Many transcriptional regulatory complexes are dependent upon multiple cooperative interactions for complex formation and maintenance. As a result, DNA-binding transcription factors are required to recruit coactivator complexes to specific DNA binding sites and the coactivator complexes are in turn necessary for stable chromatin occupancy by DNA-binding transcription factors. This effect of coactivators on DNA binding is mediated either by protein-protein interactions that stabilize protein-DNA interactions or, in the case of chromatin modifying enzymes, by increasing access of the transcription factor to its DNA binding site. Examples include the following: (a) knockdown of any member of a coactivator complex containing S100A10, ANXA2, and SPT6 inhibits the binding of OCT4, as well as the remaining members of the complex, to the *NANOG, SOX2,* and *KLF4* genes[67]; (b) HIFs recruit peptidylarginine deiminase 4 (PADI4) to HREs in a HIF-dependent manner and PADI4 recruitment is in turn required for histone citrullination and stable occupancy of HREs by HIFs[68]; (c) HIF-1 recruits the histone demethylase JMJD2C/KDM4C to HREs and JMJD2C recruitment is in turn required for histone demethylation and stable occupancy of HREs by HIF-1[9]; (d) HIF-1 recruits pyruvate kinase M2 to HREs and PKM2 enhances chromatin occupancy of HREs by HIF-1[69]; and (e) HIF-1 recruits NANOG to HREs of the *TERT* gene and NANOG recruitment is in turn required for HIF-1 protein stability and transcriptional activity[70]. In all such cases, as well as the results of the current study, it is the specificity of the DNA binding transcription factor (e.g., HIF-1) that determines where (e.g., at HREs) and when (in response to hypoxia) complex formation occurs.

In many types of human cancer, increased expression of HIF-1α[71], DNA-PKcs[54], or TRIM28[72] is associated with increased patient mortality. Several DNA-PK inhibitors are currently in clinical trials as anti-cancer agents[54]. In mouse models, inhibition of HIF transcriptional activity has significant effects on tumor vascularization, cancer stem cell specification, metastasis, and immune evasion[2,73], suggesting that the potential therapeutic effects of DNA-PK inhibitors in breast cancer may be due in part to their inhibition of HIF transcriptional activity.

## Methods

**Cell culture.** MDA-MB-231 cells were maintained in Dulbecco's modified Eagle medium (DMEM). SUM159 cells were maintained DMEM/F12 (50:50) medium. The culture media were supplemented with 10% (vol/vol) fetal bovine serum (FBS) and 1% (vol/vol) penicillin-streptomycin. Cells were cultured at 37 °C in a 5% CO$_2$, 95% air incubator. Inhibitors (NU7441 and KU55933) were dissolved in DMSO at 1000× final concentration, such that inhibitor- or vehicle-treated cells were exposed to 0.1% (vol/vol) DMSO. Cells were subjected to hypoxia in a modular incubator chamber (Billups-Rothenberg) that was flushed with a gas mixture of 1% O$_2$, 5% CO$_2$, and 94% N$_2$. Breast cancer cell lines were mycoplasma-free and authenticated by short tandem repeat DNA profiling analysis by the Johns Hopkins Genetics Resources Core Facility.

**RIME.** The procedure was performed as previously described[21]. MDA-MB-231 cells ($2 \times 10^7$) were treated with DMSO or DMOG (1 mM) for 8 h and cross-linked with 1% formaldehyde for 8 min and quenched by 0.125 M glycine. Cells were harvested and the nuclear fraction was extracted. After sonication, the nuclear lysate was immunoprecipitated with bead-prebound HIF-1α antibody (NB100-479, Novus Biologicals). Precipitated proteins in 30 µl of 100 mM ammonium hydrogen carbonate were reduced in 2.5 mM dithiothreitol at 60 °C for 1 h, then alkylated with 5 mM iodoacetamide in the dark at room temperature for 15 min. Proteins were digested with 20 ng/µl Trypsin/LysC (Promega) at 37 °C overnight. Peptides were desalted on Oasis HLB µ-elution plates (Waters), eluted with 65% acetonitrile/0.1% trifluoroacetic acid, dried by vacuum centrifugation, and analyzed by liquid chromatography/tandem mass spectrometry using a nano-Easy LC 1000 interfaced with an Orbitrap Fusion Lumos Tribrid Mass Spectrometer (Thermo Fisher Scientific). Tandem mass spectra were extracted by Proteome Discoverer version 2.3 (Thermo Fisher Scientific) and searched against the SwissProt_Full_Synchronized_2018_08 database using Mascot version 2.6.2 (Matrix Science). Mascot ".dat" files were compiled in Scaffold version 3 (Proteome Software) to validate MS/MS-based peptide and protein identifications. Peptide identifications were accepted if false discovery rate (FDR) was less than 1%, based on a concatenated decoy database search by the Peptide Prophet algorithm with Scaffold delta-mass correction[74,75].

**ChIP-qPCR assays.** ChIP was performed according to EMD Millipore ChIP assay kit (catalog # 17-295) with some modifications. Briefly, MDA-MB-231 and SUM159 cells were cross-linked in 1% formaldehyde for 10 min, quenched in 0.125 M glycine for 5 min, and lysed with SDS lysis buffer (1% SDS, 10 mM EDTA, 50 mM Tris [pH 8.1]). Chromatin was sheared by sonication to between 200 and 1000 bp. Lysates were diluted 1:10 with ChIP dilution buffer (0.01% SDS, 1.1% Triton X100, 1.2 mM EDTA, 16.7 mM Tris-HCl [pH 8.1], 167 mM NaCl), pre-cleared with salmon sperm DNA/protein A-agarose slurry (MilliporeSigma) at 4 °C for 1 h, and incubated with antibody (Supplementary Table 1) in the presence of agarose beads overnight. After sequential washes with low-salt (0.1% SDS, 1% Triton X-100, 2 mM EDTA, 20 mM Tris-HCl [pH 8.1], 150 mM NaCl), high-salt (0.1% SDS, 1% Triton X-100, 2 mM EDTA, 20 mM Tris-HCl [pH 8.1], 500 mM NaCl), LiCl (0.25 M LiCl, 1% IGEPAL-CA630, 1% deoxycholic acid, 1 mM EDTA, 10 mM Tris [pH 8.1]), and Tris-EDTA (10 mM Tris-HCl [pH 8.0], 1 mM EDTA) buffers, DNA was eluted in elution buffer (1% SDS, 0.1 M NaHCO$_3$), and cross-links were reversed by addition of 0.2 M NaCl at 65 °C for 4 h. Protein was digested with 20 µg proteinase K (Roche) for 1 h at 45 °C. DNA was purified by phenol–chloroform extraction and ethanol precipitation, and analyzed by SYBR Green-based qPCR. Fold enrichment (E) of chromatin occupancy based on immunoprecipitation (IP) from cells exposed to 1% O$_2$, as compared to cells exposed to 20% O$_2$, was calculated based on the cycle threshold (Ct) as $E = 2^{-\Delta(\Delta Ct)}$, where $\Delta Ct = Ct_{IP} - Ct_{Input}$ and $\Delta(\Delta Ct) = \Delta Ct_{IP(1\%)} - \Delta Ct_{IP(20\%)}$. Primer sequences for qPCR are shown in Supplementary Table 2.

**Co-IP assays.** Cells were lysed in immunoprecipitation lysis buffer (150 mM NaCl, 50 mM HEPES [pH 7.9], 1 mM EDTA, 10% glycerol, 1% IGEPAL, 1 mM PMSF, and 1× complete protease inhibitor cocktail), incubated 20 min on ice, and debris was cleared by centrifugation at $18,000 \times g$ at 4 °C. Lysates were then precleared by incubating with protein G-Sepharose (Amersham Biosciences) for 1 h with rotation. A 50-µg aliquot was removed for input analysis, and 1-mg aliquots of lysates were incubated with rotation overnight at 4 °C with 2 µg of the indicated antibody (Supplementary Table 3). The resulting immunoprecipitate was washed with immunoprecipitation lysis buffer four times for 5 min at 4 °C with rotation. Immunoprecipitates were subject to immunoblot assay using the indicated antibody (Supplementary Table 4).

**Immunoblot assays.** Cultured cells were lysed in RIPA buffer (Thermo Fisher Scientific) and proteins were fractionated by SDS-PAGE, blotted onto nitrocellulose membranes, which were blocked with 5% (w/v) nonfat milk in PBST for 1 h at room temperature, probed with primary antibody (Supplementary Table 4) in blocking solution overnight at 4 °C, incubated with horseradish peroxidase-conjugated secondary antibody (GE Healthcare) for 1 h at room temperature and the chemiluminescent signal was detected using ECL Plus (GE Healthcare).

**Lentiviral transduction.** Lentiviral vectors encoding shRNA targeting HIF-1α and HIF-2α were described previously[24]. pLKO.1-puro lentiviral shuttle vectors encoding shRNA targeting TRIM28, DNA-PKcs, KU70, and KU86 (MilliporeSigma; sequences are shown in Supplementary Table 5) were transfected into HEK293T cells for packaging[24]. MDA-MB-231 and SUM159 cells were transduced with viral supernatant for 24 h and then selected for stably transfected clones with puromycin (MilliporeSigma; 0.5 µg/ml for MDA-MB-231 and 1 µg/ml for SUM159). Cells were maintained in puromycin-containing medium.

**RNA-seq.** SUM159 subclones were seeded into 6-well plates in three biological replicates and exposed to 20% or 1% O$_2$ for 24 h. Total RNA was isolated using TRIzol (Invitrogen) and treated with DNase (Qiagen). Library preparation and sequencing using the NovaSeq 6000 platform (Illumina) were performed by the Johns Hopkins Genetics Resources Core Facility High-Throughput Sequencing Center. RNA-seq data were processed and interpreted using Genialis Expressions software (https://www.genialis.com). The automated data analysis on the Genialis

platform consisted of the following steps: sequence quality checks were performed on raw and trimmed reads (FastQC), trimming and quality filtering of reads (BBDuk), mapping to reference human genome Ensembl v.92 (STAR), expression quantification (featureCounts), and expression normalisation (RNA.norm). Key QC metrics (e.g., mapping statistics) were collected. As an additional quality control step, a sample of one million reads (Seqtk tool) was mapped (STAR) separately to human rRNA and globin sequences to evaluate the proportion of these reads in the sample. Differential gene expression analyses were performed with DESeq2. Lowly-expressed genes, which have expression count summed over all samples below 10, were filtered out from the differential expression analysis input matrix. Differential expression results (1% vs 20% $O_2$) with FDR < 0.05 and mRNA fold change (FC) > 1.5 were used as cutoff for identification of hypoxia-induced and hypoxia-repressed genes. We also compared different subclones (KD vs NTC) under hypoxic conditions to identify significantly changed mRNA expression (FC > 1 and FDR < 0.05) as HIF/DNA-PKcs/TRIM28-dependent genes. Finally, the combined list of mRNAs identified by these two strategies was used for Venn and Gene ontology analysis. Data were analyzed and visualized using Heatmapper for heatmaps, BioVenn for Venn diagrams, and GOnet for gene ontology[76–78].

**Public RNA-seq data analysis.** Gene set enrichment analysis (GSEA) was performed using patient data from The Cancer Genome Atlas (TCGA) breast cancer data set[79,80]. Patients were ranked by TRIM28 mRNA expression level, and the top 200 highest and lowest TRIM28-expressing patients were selected for analysis by GSEA.

**RT-qPCR.** Total RNA was isolated from cultured cells using Trizol (Invitrogen) and reverse-transcribed using the High-Capacity RNA-to-cDNA Kit (Applied Biosystems). qPCR analysis was performed using SYBR Green and the CFX96 Real-Time PCR detection system (Bio-Rad). The expression (E) of each target mRNA relative to 18S rRNA was calculated based on the cycle threshold (Ct) as $E = 2^{-\Delta(\Delta Ct)}$, in which $\Delta Ct = Ct\ (target) - Ct\ (18S)$, and $\Delta\ (\Delta Ct) = \Delta Ct\ (test\ sample) - \Delta Ct\ (control\ sample)$. PCR primer sequences are shown in Supplementary Table 6.

**Luciferase reporter assays.** SUM159 cells were seeded onto 6-well plates and transfected with the indicated plasmids, and exposed to 20% or 1% $O_2$ for 24 h. For p2.1 assay, cells were co-transfected with p2.1 and pSV-Renilla. For GalA assays, cells were co-transfected with pG5E1bLuc and either pGalA or pGalO. The FLuc/RLuc activities were measured using the Dual Luciferase Reporter Assay System (Promega).

**Metabolic assays.** Extracellular lactate was measured in conditioned medium with a lactate assay kit (MilliporeSigma). OCR was measured using a cartridge containing an optical fluorescent $O_2$ sensor in a Seahorse Bioscience XF96 Extracellular Flux Analyzer (Agilent).

**Statistics and reproducibility.** GraphPad Prism 8 software (GraphPad Software, Inc.) was used to perform statistical analysis. Two-tailed Student's $t$ test was used to test differences between two groups and differences between multiple groups were analyzed by two-way ANOVA. Data are shown as mean ± SEM of biological triplicates. Details of the statistical analysis for each experiment was indicated in the relevant figure legends. For all statistical tests, $p$ values < 0.05 were considered statistically significant. All experiments except RIME were repeated at least three times with similar results.

**Reporting summary.** Further information on research design is available in the Nature Research Reporting Summary linked to this article.

## Data availability
The data that support this study are available from the corresponding author upon reasonable request. The RNA sequencing data generated in this study have been deposited in the Gene Expression Omnibus (GEO) under accession code GSE167956. The mass spectrometry proteomics data have been deposited to the ProteomeXchange Consortium with the data set identifier PXD024373. Public TCGA data generated by the TCGA Research Network are available at https://www.cancer.gov/tcga. Source data are provided with this paper.

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

## Acknowledgements

We thank Laura Kasch-Semenza and David Mohr for library construction and RNA sequencing. We thank Rachel Geisler and Samantha Garcia (Novus Biologicals Inc.) for generously providing antibodies listed in Supplementary Tables 1, 3, and 4. We thank Dr. Pai-Sheng Chen (National Cheng Kung University, Taiwan) for providing HIF-1α deletion mutants and Dr. Ting Liu (Johns Hopkins University) for assistance with Seahorse assays. G.L.S. is an American Cancer Society Research Professor and the C. Michael Armstrong Professor at the Johns Hopkins University School of Medicine. This work was supported by grants from the American Cancer Society, Armstrong Family Foundation, and the Cindy Rosencrans Fund for Triple-Negative Breast Cancer.

## Author contributions

Y.Y. and G.L.S. conceived and designed the research study. Y.Y. performed most of the experiments and acquired data. H.L., C.C., Y.L., and R.C. helped with the experiments and provided technical advice. Y.Y. and G.L.S. analyzed the data and wrote the manuscript. G.L.S. supervised the study. All authors reviewed and commented on the manuscript.

## Competing interests

The authors declare no competing interests.
