## [Peer Review File · Nature Communications]

REVIEWER COMMENTS

Reviewer #1 (Remarks to the Author):

The manuscript titled "HIF-1 recruits TRIM28 and DNA-PK to release paused RNA polymerase II and activate target gene transcription in response to hypoxia" by Yang et al. reports that HIF-1-mediated hypoxic gene expression involves TRIM28 and DNA-PK. In this mechanism, the authors suggest that HIF-1 recruits TRIM28 and DNA-PK to hypoxia-inducible genes, DNA-PK is activated (autophosphorylation) to phosphorylate TRIM28 at S824, and phosphorylated TRIM28 at S824 recruits CDK9 to release paused Pol II. It is an interesting and potentially important finding. However, the suggested critical finding seems to be based on limited experimental data and thus requires further validation. Some of the information given in the manuscript can be also misleading, to be corrected. I am also wondering why a very limited set of hypoxic genes were investigated and in the breast cancer cells with impaired DNA damage response.

Major comments:

Only two genes ANGPTL4 and PDK1, not many, even a few hypoxia genes, are shown throughout the manuscript to argue the potentially important mechanism- TRIM28, DNA-PKcs recruitment and their activation (phosphorylation) requirement in hypoxic gene activation. The KD of DNA-PKcs and TRIM28 decreased over 1000 hypoxic gene expression, how many these genes are bound by TRIM28 and involve the phosphorylation of TRIM28 upon transcriptional activation?

The authors showed that HIF1a is not recruited to the hypoxia genes in TRIM28 KD and DNA-PK KD cells in Fig. 3E. Could it indicate that TRIM28 recruits HIF1a to the activated genes? In the text, the authors argue that TRIM28 is recruited by HIF1a upon the gene activation. Although the authors showed TRIM28 recruitment on two hypoxia genes by ChIP-qPCR in Fig. 1, multiple previous papers reported that a large number of genes are occupied by TRIM28 even before transcriptional activation. In fact, TRIM28 appears to be associated with DNA as some reported a consensus TRIM28 binding sequence. Moreover, DNA-PKcs is one of the most abundant proteins ubiquitously associated with the genome. Have the authors tested whether TRIM28 and DNA-PKcs are recruited to the hypoxia-inducible genes in HIF1a KD cells? This experiment should be performed to validate the role of these transcription factors and the sequence of transcriptional activation events.

In Fig. 4H, DNA-PKcs phosphorylation at T2609 was inhibited by NU7441. This residue T2609, as well as TRIM28 S824, was reported to be ATM-dependent by multiple papers (DOI: 10.1128/MCB.00048-06; DOI: 10.1042/BJ20020973), which should be stated in the manuscript. In addition, ATM inhibitor should be tested to determine whether ATM is not involved in the phosphorylation of DNA-PKcs and TRIM28. In addition to immunoblotting, ChIP-qPCR results showing DNA-PKcs T2609 and TRIM28 S824 at some hypoxia target genes are to be shown in WT and ATM- and DNA-PK-inhibited cells.

In addition, it is well-known that ATM (as well as ATR) is activated during hypoxia as multiple papers have shown the importance of ATM in the expression of hypoxia-activated genes. The authors' argument that DNA damage and H2AX phosphorylation occur only at near-anoxic oxygen levels is also contradictory to the previous finding (e.g. "Histone H2AX is integral to hypoxia-driven neovascularization" by Economopoulou et al. Nat. Med. 2009—in which a similar oxygen level was applied, to the one in this study). The authors argument that ATM activity is not required for HIF target gene expression needs to be more thoroughly validated because it is contradictory to previous reports. This should also be discussed in the manuscript.

It seems possible that the DNA damage response signaling in the current study could be affected by the massive p53 mutations in MDA-MB-231 cells.

In page 12, the claim "residue 824 of TRIM28 (by DNA-PK-dependent serine phosphorylation or

aspartate substitution)" needs to be corrected because TRIM28 S824 is a substrate of ATM and DNA-PK. In fact, TRIM28 S824 is more known as an early substrate of ATM during DNA damage.

Importantly, it is more likely that HIF1a is not pulled down with TRIM28 in normal cell extracts without hypoxic stress in Fig. 1D and 1E because HIF1a protein is present in hypoxic condition as shown in Fig. 1D. The current description is easy to be misunderstood or mislead as TRIM28 can interact with HIF1a only in hypoxia.

In Fig. 7B, HA pull-down is to be shown.

The IP assays described in the method section don't include any wash steps, which concerns of unspecific or loose interactions sustained to be shown in figures. Overall, the method description is insufficient and should be more thorough.

Similarly, CDK9 recruitment by phosphorylated TRIM28 requires further validation. P-TEFb is known to be recruited by Mediator complex, gene-specific transcription factors etc. In the current manuscript, it is unclear whether the interaction between CDK9 and phosphorylated TRIM28 (S824) is direct or indirect. The stringency of IP is also questionable. Can endogenous CDK9 pull-down endogenous phosphorylated TRIM28 at S824 upon hypoxia and vice versa?

Minor comments:

In Fig. G, the description for each lane on the blots is missing.

In Fig. 3E, the two sets of three bars of DNA-PKcs and TRIM28, HIF1a, HIF1b, HIF2a for ANGPTL4 20 appear too identical?

What is the p-value and TPM cut off for the 1307 upregulated genes in RNA-seq? These are to be stated in page 8.

Reviewer #2 (Remarks to the Author):

SUMMARY

In this report the authors identify TRIM28 and DNA-PK as co-regulators of HIF1 transcriptional targets. Although both have been previously implicated in transcriptional control, this report is the first to describe their role in transcription regulation by HIFs.

Starting from a previously published dataset identifying TRIM28 (aka KAP1) and DNA-PK subunits as putative interactors of chromatin-bound HIF1, the authors validate these factors as both physical and functional interactors, demonstrating the formation of a complex containing these factors and their recruitment to chromatin at hypoxia-inducible genes in a HIF1-dependent manner. Subsequent experiments demonstrate that TRIM28 and DNA-PK are required for full induction of hypoxia-inducible genes and that the kinase activity of DNA-PK is also necessary. Evidence is presented in support of a model where TRIM28 and DNA-PK contribute to the HIF-dependent stimulation of CDK9 recruitment and release of paused RNAPII during hypoxia.

Overall, the experiments and findings described here are largely sound and well-supported, with the exception of the points detailed below. The manuscript and the HIF/hypoxia field would greatly benefit from additional mechanistic insights such as determining the influence of TRIM28 and DNA-PK on other known HIF1 coactivators such as Mediator/CDK8 and/or whether their HIF1-dependent recruitment is direct or via association with active transcription complexes more generally.

MAJOR COMMENTS

The ChIP + qPCR experiments described are the standard practice for quantitative assessment of factor recruitment to chromatin. However, using a single amplicon around the HRE region does not provide sufficient data to assess enrichment over background or resolution. More importantly, the use of a single amplicon near the 5' end of a gene does not measure if the recruitment occurs in a defined peak region versus a more broad region of enrichment that may provide insight into whether recruitment is direct vs. via association with transcription machinery. Finally, the HRE region is not ideal for quantitative assessment of factors that are more likely to be enriched in the downstream / gene body region such as CTD S5-phosphorylated RNAPII or CDK9. The authors should also clarify if the stated N=3 represents distinct biological replicates of the entire ChIP experiment vs. replication of the qPCR only.

Co-IP assays as described do not formally test if the interactions occur "in the absence of chromatin", unless further steps such as DNA degradation are performed. A more thorough description of the co-IP methodology should be provided, and it would also be helpful to include stained gel images to assess the background vs. specificity of the pulldowns.

A more thorough description of the RNA-seq methodology is required. What were the criteria for QC and/or quality filtering? How many reads total were obtained vs. aligned? Most importantly, there is no description of the statistical approach used to assess differential expression or the FDR correction method used. In addition, while qualitative threshold-based comparison of the pair-wise hypoxia/normoxia changes in the different cell lines, as in the venn diagram, can be informative, this does not constitute an effective statistical test for differences in hypoxia-driven changes. At the very least one should include comparisons of the KD/control in hypoxia to test for differences in the induced expression levels, but better would be to test for an interaction between treatment (hypoxia) and KD ie are there differences in the hypoxia-driven changes between the control and KD? How global is the effect of the KDs vs specific to hypoxia-inducible genes? Inclusion as a supplementary panel(s) of MA and/or volcano plots for each of the comparisons to assess the variability and magnitude of changes would also be helpful. Finally, unless I missed this, there is no mention of this data set being made available, eg in NCBI Gene Expression Omnibus.

The statement regarding HIF-dependent repression would be better supported by showing that some of the known repressors directly induced by HIF1A are indeed reduced in the KDs. It would also be helpful to assess enrichment of transcription factor targets, eg using the transcription factor targets module in the Metascape tool or similar.

MINOR COMMENTS

"recent studies indicate that for 30% of all protein-coding genes, and 70% of those induced by a physiological stimulus such as heat shock or hypoxia, Pol II binds to the promoter and initiates transcription, but pauses 20-100 nucleotides downstream of the transcription start site, as a result of its interaction with negative elongation factor (NELF) and DRB-sensitivity inducing factor (DSIF) [16-18]."

This should clarify that pausing occurs at ALL protein-coding genes, while being a rate-limiting step at 40-70%, depending on cell type/method), especially stimulus-responsive genes such as those induced by heatshock or hypoxia.

Estimates of pausing location have varied but the latest consensus (and the cited reference) is ~ 30-60 nt downstream of the TSS.

"positive transcription elongation factor b (P-TEFb), consisting of CDK9 and cyclin T1, is recruited to paused Pol II"

Would be helpful to add “through a variety of mechanisms” here plus references.

In the section covering background knowledge of TRIM28 and in the discussion, it should be noted that TRIM28 can itself directly bind chromatin (Bacon et al. 2020; PMID: 32402252) and the implications of this for CDK9 recruitment and the current work should be discussed.

In their introduction of the RIME dataset, the authors may wish to add that Pontin and Reptin have been identified as HIF interactors in *Drosophila* (Dekanty et al. 2010; 20585616).

A brief description of RT-qPCR normalization should be included in figure legends, rather than just methods.

Arrangement of panels in Fig 3C is potentially confusing.

“also inhibited by NU7441 in a dose dependent manner (Fig. 4I).”

Cannot really conclude dose-dependency with only two concentrations – suggest reword to “at two doses”

Knockdown with rescue experiments in Fig 5E are a great approach, but it should at least be discussed/indicated how the expression levels compare to endogenous TRIM28 expression. As shown here in only the KD background it is impossible to judge. Furthermore, in this context the statement that “TRIM28 is necessary and sufficient for activation of HIF target gene expression” is a bit strong and should be toned down.

“whereas S2 phosphorylation (S2P) triggers release” should be reworded to reflect that the CTD phosphorylations have not been formally shown to trigger different steps in the transcription cycle, but rather are highly correlated/associated with certain stages.

“indicating that TRIM28 and DNA-PK are required for S2P of paused Pol II in response to hypoxia.” This statement is somewhat too strong given that ChIP does not specifically measure transcriptional activity. Suggest rewording along the lines of “is consistent with...”

“subsequent productive transcriptional elongation require the phosphorylation and dissociation of NELF” should replace with “requires”

In Fig 6C how are the relative levels at each region measured/presented? It seems odd that the levels at the 5' region should be so much lower than the downstream regions. Please also clarify if the cDNA protocol used oligo dT and/or random hexameric primers as this may help with interpretation.

“by contrast, Pol II pausing (Fig. 6A) and transcriptional elongation (Fig. 5F) at HIF target genes under non-hypoxic conditions were not altered in TRIM28-KD breast cancer cells.” This statement is not robustly addressed by the current analysis. Suggest tone down or rephrasing.

Please check for full descriptions of all methods – for example full recipes should be given for all ChIP buffers, incubation time with antibodies is missing, and reverse cross-linking usually includes elevated temperature.

Reviewer #3 (Remarks to the Author):

This submission by Yang et al describes an interesting study of the roles of TRIM 28 and the enzyme DNA-PK in the transcriptional response to hypoxia mediated by hypoxia inducible factor. The experiments appear to have been carried out very thoroughly and explore multiple HIF family

members, a range of hypoxia-inducible genes and a number of cell lines and after some minor additions should be published.

Previous studies have suggested role for programmed DNA strand breaks and the DNA damage response in transcription of heat shock, serum-induced and nuclear receptor mediated transcription (Madabhushi, R. Cell 2015, 161, 1592 and Bunch et al, Nat Commun, 2015: 6, 10191). The DSB were regarded as a direct product of increased trans activation rather than an effect of the treatment used and thought to be generated by topoisomerase 2. DNA-PK was thus thought of part of the non-homologous end joining response to DSB generation during transcription. The authors here suggest a modification of this mechanism in transcriptional responses to hypoxia in which DNA-PK and its substrate TRIM28 are recruited by HIF-1 to the promoter independently of DSB. While this is an interesting and compelling mechanism, it would be good to have more evidence for the lack of involvement of topo2 and DSB in the transcriptional function of HIF1. The experiments in Fig. 4 G attempt to address this, although g-H2Ax levels are examined after eight hours hypoxia. Earlier points should be looked at as transcription is likely induced much earlier, likely within minutes. In addition, CHIP-PCR assays of g-H2Ax association with the hypoxia-induced genes during hypoxia should be carried out. It would also be recommended to try to exclude a role for topo2 in hypoxia-inducible transcription using topo2 inhibitors.

Point-by-Point Response to Reviewers' Comments

We appreciate the reviewers' thoughtful comments and critiques. As suggested, we have performed extensive new experiments and extensive revision of the manuscript, which now contains 8 figures, 8 supplementary figures and 7 supplementary tables. We have also added many statements of clarification to the Methods and Discussion sections. Text revisions are shown in red in the revised manuscript. Below, we describe, point-by-point, the changes that were made in the manuscript to address each of the reviewer's comments.

Reviewer #1

The manuscript titled "HIF-1 recruits TRIM28 and DNA-PK to release paused RNA polymerase II and activate target gene transcription in response to hypoxia" by Yang et al. reports that HIF-1-mediated hypoxic gene expression involves TRIM28 and DNA-PK. In this mechanism, the authors suggest that HIF-1 recruits TRIM28 and DNA-PK to hypoxia-inducible genes, DNA-PK is activated (autophosphorylation) to phosphorylate TRIM28 at S824, and phosphorylated TRIM28 at S824 recruits CDK9 to release paused Pol II. It is an interesting and potentially important finding. However, the suggested critical finding seems to be based on limited experimental data and thus requires further validation. Some of the information given in the manuscript can be also misleading, to be corrected. I am also wondering why a very limited set of hypoxic genes were investigated and in the breast cancer cells with impaired DNA damage response.

Major comments:

1. Only two genes *ANGPTL4* and *PDK1*, not many, even a few hypoxia genes, are shown throughout the manuscript to argue the potentially important mechanism- *TRIM28*, *DNA-PKcs* recruitment and their activation (phosphorylation) requirement in hypoxic gene activation. The KD of *DNA-PKcs* and *TRIM28* decreased over 1000 hypoxic gene expression, how many these genes are bound by *TRIM28* and involve the phosphorylation of *TRIM28* upon transcriptional activation?

Response: We have now analyzed 8 additional HIF target genes (*LOX*, *LDHA*, *CA9*, *VEGFA*, *PKM2*, *SLC2A3*, *PGK1* and *PGF*; at left). ChIP-qPCR assays revealed that, as in the case of *ANGPTL4* and *PDK1*, the HRE of each of these 8 HIF target genes was occupied by *DNA-PK* and *TRIM28* in response to hypoxia, indicating that for 10 out of 10 hypoxia-

induced genes analyzed, DNA-PK and TRIM28 were recruited in response to hypoxia. **These data are presented as Supplementary Fig. 1B in the revised manuscript.** Based on these results, RNA-seq data, and extensive additional data presented in the manuscript, we conclude that TRIM28 and DNA-PK are required for HIF target gene transcription in breast cancer cells.

2. *The authors showed that HIF1 α is not recruited to the hypoxia genes in TRIM28 KD and DNA-PK KD cells in Fig. 3E. Could it indicate that TRIM28 recruits HIF1 α to the activated genes? In the text, the authors argue that TRIM28 is recruited by HIF1 α upon the gene activation. Although the authors showed TRIM28 recruitment on two hypoxia genes by ChIP-qPCR in Fig. 1, multiple previous papers reported that a large number of genes are occupied by TRIM28 even before transcriptional activation. In fact, TRIM28 appears to be associated with DNA as some reported a consensus TRIM28 binding sequence. Moreover, DNA-PKcs is one of the most abundant proteins ubiquitously associated with the genome. Have the authors tested whether TRIM28 and DNA-PKcs are recruited to the hypoxia-inducible genes in HIF1 α KD cells? This experiment should be performed to validate the role of these transcription factors and the sequence of transcriptional activation events.*

Response: To address this comment, we performed ChIP-qPCR to analyze SUM159 subclones expressing a non-targeting control (NTC) shRNA or shRNA targeting HIF-1 α . Hypoxia-induced recruitment of TRIM28 and DNA-PKcs was significantly decreased in cells expressing shRNA targeting HIF-1 α (see below). **These data are presented as Supplementary Fig. 2 in the revised manuscript.** We also demonstrated that formation of the DNA-PK/TRIM28 complex is dependent on HIF expression (Fig. 1I). Finally, HIFs bind to the 5'-RCGTG-3' sequence present in all HREs, thereby providing the sequence specificity for complex formation and transcriptional activation.

3. *In Fig. 4H, DNA-PKcs phosphorylation at T2609 was inhibited by NU7441. This residue T2609, as well as TRIM28 S824, was reported to be ATM-dependent by multiple papers (DOI: 10.1128/MCB.00048-06; DOI: 10.1042/BJ20020973), which should be stated in the manuscript. In addition, ATM inhibitor should be tested to determine whether ATM is not involved in the phosphorylation of DNA-PKcs and TRIM28. In addition to immunoblotting, ChIP-qPCR results*

showing DNA-PKcs T2609 and TRIM28 S824 at some hypoxia target genes are to be shown in WT and ATM- and DNA-PK-inhibited cells.

Response: We performed two additional experiments as suggested by the reviewer. First, we exposed SUM159 cells to 20% or 1% O₂ for 8 h, in the presence of DMSO, NU7441 (4 μ M) or KU55933 (20 μ M) and performed immunoblot assays. The DNA-PK inhibitor NU7441, but not the ATM inhibitor Ku55933, decreased the

phosphorylation of DNA-PKcs at T2609 and phosphorylation of TRIM28 at S824. These data are presented as Supplementary Fig. 6D in the revised manuscript.

Second, we performed ChIP-qPCR assay in SUM159 cells. The cells were exposed to 20% or 1% O₂ for 16 hours, in the presence of DMSO or NU7441 (4 μM) or KU55933 (20 μM) and ChIP assays were performed using antibody against phospho-S824-TRIM28 or phospho-T2609-DNA-PKcs. The data demonstrate that the DNA-PK inhibitor, NU7441, but not ATM inhibitor, Ku55933 blocked occupancy of the *ANGPTL4* and *PDK1* HREs by phospho-S824-TRIM28 or phospho-T2609-DNA-PKcs. These data are presented as Supplementary Fig. 6E in the revised manuscript.

4. In addition, it is well-known that ATM (as well as ATR) is activated during hypoxia as multiple papers have shown the importance of ATM in the expression of hypoxia-activated genes. The authors' argument that DNA damage and H2AX phosphorylation occur only at near-anoxic oxygen levels is also contradictory to the previous finding (e.g. "Histone H2AX is integral to hypoxia-driven neovascularization" by Economopoulou et al. *Nat. Med.* 2009—in which a similar oxygen level was applied, to the one in this study). The authors' argument that ATM activity is not required for HIF target gene expression needs to be more thoroughly validated because it is contradictory to previous reports. This should also be discussed in the manuscript.

Response: Multiple publications have documented that in many cell types hypoxia-induced DNA damage occurs only at very low oxygen levels (< 0.1% O₂) and that 1% O₂ does not induce detectable DNA damage (e.g., Hammond et al., *J Biol Chem.* 2003;278:12207; Bencokova et al., *Mol Cell Biol.* 2009;29:526; Wrann et al., *Biol Chem.* 2013;394:519). However, Economopoulou et al. (*Nat Med.* 2009;15:553) have reported that γH2AX was induced by exposure of human umbilical vein endothelial cells to 1% O₂ in an ATR-dependent manner, which was required to maintain proliferation and hypoxia-induced neovascularization. An explanation for this apparent inconsistency is that responses to hypoxia utilize different signaling mechanisms in different cell types. Our data from breast cancer cells are consistent with other reports in the literature (e.g. Bouquet et al., *J Cell Sci.* 2011;124:1943; Venkatesh et al., *Oncoimmunology.* 2020; 9:1750750). We have cited these papers in the Discussion section on page 19 of the revised manuscript.

5. It seems possible that the DNA damage response signaling in the current study could be affected by the massive p53 mutations in MDA-MB-231 cells.

Response: It is known that MDA-MB-231 cells overexpress mutant p53, whereas SUM-159 cells do not express p53 (Olivier et al., *Hum Mut.* 2002; 19:607; Neve et al., *Cancer Cell* 2006;10:515), and constitutive activation of DNA damage checkpoint signaling contributes to mutant p53 accumulation (Frum et al., *Mol Cancer Res.* 2016;14:423). However, without exception, our data demonstrate that exposure of breast cancer cells to 1% O₂ treatment does not cause DNA damage and that the recruitment/activation of DNA-PK at HREs is independent of DNA damage response

signaling and is instead dependent on interaction with HIFs and TRIM28. We have highlighted this finding by creating a separate heading above these data on page 11 of the revised manuscript that reads: “DNA damage and ATM activation are not induced by exposure of breast cancer cells to 1% O₂.” We have included extensive new experimental data in Supplementary Figure 6 showing that there is no evidence of ATM or H2AX phosphorylation or any requirement for ATM in the induction of HIF target genes in hypoxic breast cancer cells (see Supplementary Figures 6D and 6E, which are presented in response to comment 3, above).

6. *In page 12, the claim “residue 824 of TRIM28 (by DNA-PK-dependent serine phosphorylation or aspartate substitution)” needs to be corrected because TRIM28 S824 is a substrate of ATM and DNA-PK. In fact, TRIM28 S824 is more known as an early substrate of ATM during DNA damage.*

Response: It is true that TRIM28 is a substrate of ATM during DNA damage. However, our results indicate that there is no DNA damage induced by exposure of breast cancer cells to 1% O₂. We present extensive experimental data demonstrating that hypoxia-induced phosphorylation of TRIM28 on S824 was dependent on DNA-PK and HIFs and was independent of ATM (Fig. 4G and Supplementary Fig. 6C-E).

7. *Importantly, it is more likely that HIF1a is not pulled down with TRIM28 in normal cell extracts without hypoxic stress in Fig. 1D and 1E because HIF1a protein is present in hypoxic condition as shown in Fig. 1D. The current description is easy to be misunderstood or mislead as TRIM28 can interact with HIF1a only in hypoxia.*

Response: We thank the reviewer for pointing out this potential source of misunderstanding. In an attempt to clarify this issue, we have inserted the following statement at the very beginning of the Discussion section on page 18 of the revised manuscript: “Under hypoxic conditions, O₂-dependent hydroxylation, ubiquitination and degradation of HIF-1α is inhibited, leading to rapid protein accumulation [2, 3, 6, 7]. The data presented in this study demonstrate that the induction of HIF-1α protein expression in response to hypoxia leads to interaction with TRIM28. Together, HIF-1α and TRIM28 assemble the DNA-PK heterotrimer, which autophosphorylates T2609 and phosphorylates TRIM28 on S824, enabling TRIM28 to recruit CDK9, which phosphorylates Pol II CTD on S5, triggering release of the paused polymerase, leading to productive transcriptional elongation of HIF target genes (Fig. 8D).”

8. *In Fig. 7B, HA pull-down is to be shown.*

Response: In response to the reviewer’s comment, we performed the requested HA pull down experiment and present the results in Fig. 7B (right panel) of the revised manuscript.

9. *The IP assays described in the method section don’t include any wash steps, which concerns of unspecific or loose interactions sustained to be shown in figures. Overall, the method description is insufficient and should be more thorough.*

Response: We have revised the IP assay description in the Methods section on pages 22-23 of the revised manuscript to read as follows: “**Co-IP assays.** Cells were lysed in immunoprecipitation lysis buffer (150 mM NaCl, 50 mM HEPES, pH 7.9, 1 mM EDTA, 10% glycerol, 1% IGEPAL, 1 mM PMSF, and 1 × complete protease inhibitor cocktail) and incubated 20 min on ice, and debris was cleared by centrifugation at 13,000 rpm at 4 °C. Lysates were then precleared by incubating with protein G-Sepharose (Amersham Biosciences) for 1h at 4 °C with rotation. A 50-μg aliquot was removed for input analysis, and 1-mg aliquots of lysates were incubated with 2 μg of the indicated antibodies (Supplementary Table 4) with rotation overnight at 4 °C. The resulting immunoprecipitate was washed with immunoprecipitation lysis buffer four times for 5 min at 4 °C with rotation. Immunoprecipitates were analyzed by Western blotting using the indicated antibodies (Supplementary Table 5).”

10. *Similarly, CDK9 recruitment by phosphorylated TRIM28 requires further validation. P-TEFb is known to be recruited by Mediator complex, gene-specific transcription factors etc. In the current manuscript, it is unclear whether the interaction between CDK9 and phosphorylated TRIM28 (S824) is direct or indirect. The stringency of IP is also questionable. Can endogenous CDK9 pull-down endogenous phosphorylated TRIM28 at S824 upon hypoxia and vice versa?*

Response: To address the reviewer’s concern, we performed an endogenous Co-IP assay to examine the interaction of CDK9 and phosphorylated TRIM28 at S824. We exposed the SUM159 cells to 1% O₂ for 8 h and immunoprecipitation was performed with CDK9, TRIM28 or p-S824-TRIM28 antibody. The results showed CDK9 interacts with TRIM28 and phosphorylated TRIM28 in response to hypoxia, which is consistent with the overexpression data shown in Fig. 7B. **These data are presented as Fig. 7D in the revised manuscript.**

Minor comments:

1. *In Fig. G, the description for each lane on the blots is missing.*

Response: We thank the reviewer for the comment. We have relabeled Fig. 1G and added a detailed description in the figure legend.

2. In Fig. 3E, the two sets of three bars of DNA-PKcs and TRIM28, HIF1a, HIF1b, HIF2a for ANGPTL4 20 appear too identical?

Response: For each Ab, the qPCR data have been normalized to the mean value for cells exposed to 20% O₂. We have revised the figure legend to state this explicitly.

3. What is the p-value and TPM cut off for the 1307 upregulated genes in RNA-seq? These are to be stated in page 8.

Response: We have added the following information on page 8: “Hypoxia significantly increased the expression of 1,307 mRNAs by at least 1.5-fold (FDR < 0.05) in a HIF-dependent manner (Supplementary Table 1)”. For the RNA-seq data analysis, the filter we used was based on FDR (adjusted p-value) and raw counts (not TPM). FDR is not exactly the p-value, but is derived from the p-value and the number of hypotheses tested. It is essential to consider multiple hypotheses testing for more accurate results. Raw counts are considered because those are the inputs for DESeq2. DESeq2 removes low expressing genes. Genes where the sum of expression values across all samples is less than 10 are removed. Even if one supplies DESeq2 with a normalized expression matrix, it is not advisable to remove low expressors based on TPM. Thus, we do that based on raw counts. A more thorough description of the RNA-seq methodology is also provided in the Methods section on pages 23-24 of the revised manuscript: “**RNA-seq.** SUM159 subclones were seeded into 6-well plates in three biological replicates and exposed to 20% or 1% O₂ for 24 hours. Total RNA was isolated using TRIzol (Invitrogen) and treated with DNase (Qiagen). Library preparation and sequencing using the NovaSeq6000 platform (Illumina) were performed by Johns Hopkins Genetics Resources Core Facility High-Throughput Sequencing Center. RNA-seq data were processed and interpreted using Genialis Expressions software. The automated data analysis pipeline run in the Genialis platform consisted of the following analysis steps: sequence quality checks were performed on raw and trimmed reads (FASTQC), trimming and quality filtering of reads (BBduk), mapping to a reference human genome Ensembl v.92 (STAR), expression quantification (featureCounts), and expression normalisation (rnanorm). Key QC metrics (e.g. mapping statistics) are collected. As an additional quality control step, a sample of a million reads (Seqtk tool) is mapped (STAR) separately to human rRNA and globin sequences to evaluate the proportion of these kinds of reads in the sample. Differential gene expression analyses were performed with DESeq2. Lowly-expressed genes, which have expression count summed over all samples below 10, were filtered out from the differential expression analysis input matrix (<https://www.genialis.com/>). Differential expression results with FDR < 0.05 and mRNA fold change > 1.5 were used as a cutoff for further analysis. Raw RNA sequencing data have been uploaded to GEO with identifier GSE167956 [<https://www.ncbi.nlm.nih.gov/geo/query/acc.cgi?acc=GSE167956>].”

References

1. Hammond EM, Dorie MJ, Giaccia AJ. ATR/ATM targets are phosphorylated by ATR in response to hypoxia and ATM in response to reoxygenation. *J. Biol. Chem.* **278**, 12207-12213 (2003)
2. Bencokova Z, Kaufmann MR, Pires IM, Lecane PS, Giaccia AJ, Hammond EM. ATM activation and signaling under hypoxic conditions. *Mol. Cell. Biol.* **29**, 526-537 (2009).

3. Wrann S, Kaufmann MR, Wirthner R, Stiehl DP, Wenger RH. HIF mediated and DNA damage independent histone H2AX phosphorylation in chronic hypoxia. *Biol. Chem.* **394**, 519-528 (2013).
4. Economopoulou M, et al. Histone H2AX is integral to hypoxia-driven neovascularization. *Nat. Med.* **15**, 553-558 (2009).
5. Bouquet, F. et al. A DNA-dependent stress response involving DNA-PK occurs in hypoxic cells and contributes to cellular adaptation to hypoxia. *J. Cell Sci.* **124**, 1943-1951 (2011).
6. Hassan Venkatesh G, et al. Hypoxia increases mutational load of breast cancer cells through frameshift mutations. *OncoImmunology* **9**, e1750750 (2020).
7. Olivier M, Eeles R, Hollstein M, Khan MA, Harris CC, Hainaut P. The IARC TP53 database: new online mutation analysis and recommendations to users. *Hum. Mutat.* **19**, 607-614 (2002).
8. Neve RM, et al. A collection of breast cancer cell lines for the study of functionally distinct cancer subtypes. *Cancer Cell* **10**, 515-527 (2006).
9. Frum RA, et al. Constitutive activation of DNA damage checkpoint signaling contributes to mutant p53 accumulation via modulation of p53 ubiquitination. *Mol. Cancer Res.* **14**, 423-436 (2016).

Reviewer #2

SUMMARY

In this report the authors identify TRIM28 and DNA-PK as co-regulators of HIF1 transcriptional targets. Although both have been previously implicated in transcriptional control, this report is the first to describe their role in transcription regulation by HIFs.

Starting from a previously published dataset identifying TRIM28 (aka KAP1) and DNA-PK subunits as putative interactors of chromatin-bound HIF1, the authors validate these factors as both physical and functional interactors, demonstrating the formation of a complex containing these factors and their recruitment to chromatin at hypoxia-inducible genes in a HIF1-dependent manner. Subsequent experiments demonstrate that TRIM28 and DNA-PK are required for full induction of hypoxia-inducible genes and that the kinase activity of DNA-PK is also necessary. Evidence is presented in support of a model where TRIM28 and DNA-PK contribute to the HIF-dependent stimulation of CDK9 recruitment and release of paused RNAPII during hypoxia.

Response: The RIME dataset was unpublished prior to this report.

Overall, the experiments and findings described here are largely sound and well-supported, with the exception of the points detailed below. The manuscript and the HIF/hypoxia field would greatly benefit from additional mechanistic insights such as determining the influence of TRIM28 and DNA-PK on other known HIF1 coactivators such as Mediator/CDK8 and/or whether their HIF1-dependent recruitment is direct or via association with active transcription complexes more generally.

MAJOR COMMENTS

1. *The ChIP + qPCR experiments described are the standard practice for quantitative assessment of factor recruitment to chromatin. However, using a single amplicon around the HRE region does not provide sufficient data to assess enrichment over background or resolution. More importantly, the use of a single amplicon near the 5' end of a gene does not measure if the recruitment occurs in a defined peak region versus a more broad region of enrichment that may provide insight into whether recruitment is direct vs. via association with transcription machinery. Finally, the HRE region is not ideal for quantitative assessment of factors that are more likely to be enriched in the downstream / gene body region such as CTD S5-phosphorylated RNAPII or CDK9. The authors should also clarify if the stated N=3 represents distinct biological replicates of the entire ChIP experiment vs. replication of the qPCR only.*

Response: HREs are in fact present at various locations in HIF target genes, including 5' flanking sequence, exons, introns and 3'-flanking sequence. We chose two typical HIF target genes for this study, *ANGPTL4* and *PDK1* (the location of HREs are shown below). For *ANGPTL4*, the HRE is located in the 5'-flanking region, whereas the HRE in the *PDK1* gene is located in the first intron.

HIF binding site in ANGPTL4 gene

HIF binding site in PDK1 gene

The 8 additional HIF target genes (*LOX*, *LDHA*, *CA9*, *VEGFA*, *PKM2*, *SLC2A3*, *PGK1* and *PGF*) analyzed by ChIP-qPCR in Supplementary Fig. 1B of the revised manuscript further increase the diversity of HRE locations with respect to the transcription start site. Based on these data, as well as extensive immunoprecipitation analyses of soluble (i.e. non-chromatin-associated) lysates, we are confident that the conclusions we draw regarding direct protein-protein interactions are correct. N = 3 shown here represents three biological replicates, which we now state explicitly in the revised manuscript.

2. *Co-IP assays as described do not formally test if the interactions occur “in the absence of chromatin”, unless further steps such as DNA degradation are performed. A more thorough description of the co-IP methodology should be provided, and it would also be helpful to include stained gel images to assess the background vs. specificity of the pulldowns.*

Response: We have revised the description in the Methods section on pages 22-23 of the revised manuscript to read as follows: “**Co-IP assays.** Cells were lysed in immunoprecipitation lysis buffer

(150 mM NaCl, 50 mM HEPES [pH 7.9], 1 mM EDTA, 10% glycerol, 1% IGEPAL, 1 mM PMSF, and 1 × complete protease inhibitor cocktail) and incubated 20 min on ice, and debris was cleared by centrifugation at 13,000 rpm at 4 °C. Lysates were then precleared by incubating with protein G-Sepharose (Amersham Biosciences) for 1 h at 4 °C with rotation. A 50-µg aliquot was removed for input analysis, and 1-mg aliquots of whole cell lysates were incubated with rotation overnight at 4 °C with 2 µg of the indicated antibodies (Supplementary Table 4). The resulting immunoprecipitate was washed with immunoprecipitation lysis buffer four times for 5 min at 4 °C with rotation. Immunoprecipitates were analyzed by immunoblot assay using the indicated antibodies (Supplementary Table 5).”

3. *A more thorough description of the RNA-seq methodology is required. What were the criteria for QC and/or quality filtering? How many reads total were obtained vs. aligned? Most importantly, there is no description of the statistical approach used to assess differential expression or the FDR correction method used. In addition, while qualitative threshold-based comparison of the pair-wise hypoxia/normoxia changes in the different cell lines, as in the venn diagram, can be informative, this does not constitute an effective statistical test for differences in hypoxia-driven changes. At the very least one should include comparisons of the KD/control in hypoxia to test for differences in the induced expression levels, but better would be to test for an interaction between treatment (hypoxia) and KD ie are there differences in the hypoxia-driven changes between the control and KD?*

Response: A more thorough description of the RNA-seq methodology is provided in the Methods section on pages 23-24 of the revised manuscript, which reads as follows: “**RNA-seq.** SUM159 subclones were seeded into 6-well plates in three biological replicates and exposed to 20% or 1% O₂ for 24 hours. Total RNA was isolated using TRIzol (Invitrogen) and treated with DNase (Qiagen). Library preparation and sequencing using the NovaSeq 6000 platform (Illumina) were performed by Johns Hopkins Genetics Resources Core Facility High-Throughput Sequencing Center. RNA-seq data were processed and interpreted using Genialis Expression software (<https://www.genialis.com>). The automated data analysis on the Genialis platform consisted of the following steps: sequence quality checks were performed on raw and trimmed reads (FastQC), trimming and quality filtering of reads (BBduk), mapping to a reference human genome Ensembl v.92 (STAR), expression quantification (featureCounts), and expression normalisation (RNA.norm). Key QC metrics (e.g. mapping statistics) are collected. As an additional quality control step, a sample of one million reads (Seqtk tool) is mapped (STAR) separately to human rRNA and globin sequences to evaluate the proportion of these reads in the sample. Differential gene expression analyses were performed with DESeq2. Lowly-expressed genes, which have expression count summed over all samples below 10, were filtered out from the differential expression analysis input matrix (<https://www.genialis.com>). Differential expression results with FDR < 0.05 and mRNA fold change > 1.5 were used as a cutoff for further analysis. Raw RNA sequencing data have been uploaded to GEO with identifier GSE167956 [<https://www.ncbi.nlm.nih.gov/geo/query/acc.cgi?acc=GSE167956>].”

4. How global is the effect of the KDs vs specific to hypoxia-inducible genes? Inclusion as a supplementary panel(s) of MA and/or volcano plots for each of the comparisons to assess the variability and magnitude of changes would also be helpful.

Finally, unless I missed this, there is no mention of this data set being made available, eg in NCBI Gene Expression Omnibus.

Response: We generated a volcano plot to show the differential gene expression in NTC, DNA-PKcs-KD and TRIM28-KD subclones. These data are presented as Supplementary Fig. 3E in the revised manuscript. The raw RNA sequencing data have been uploaded to GEO with identifier GSE167956 [<https://www.ncbi.nlm.nih.gov/geo/query/acc.cgi?acc=GSE167956>].

5. The statement regarding HIF-dependent repression would be better supported by showing that some of the known repressors directly induced by HIF1A are indeed reduced in the KDs. It would also be helpful to assess enrichment of transcription factor targets, eg using the transcription factor targets module in the Metascape tool or similar.

Response: We thank the reviewer for this comment. The vast majority of hypoxia-repressed genes are regulated by HIF-1 indirectly. As stated in the manuscript, HIF-1 represses these genes by activating the transcription of genes encoding transcriptional repressors or microRNAs. For example, HIF-dependent hypoxia-induced expression of miR-21 leads to decreased expression of the tumor suppressor SPRY2 (Shen *et al.*, *Acta Pharmacologica Sinica*. 2013;34(3):336-341; Sayed *et al.*, *Mol Biol Cell*. 2008;19(8):3272–3282), which is hypoxia-repressed in NTC, but not in HIF-KD, cells according to our RNA-seq data. As suggested, we performed the enrichment analysis of HIF-dependent hypoxia-repressed genes using the Metascape tool and the results are shown below. The raw RNA sequencing data have been uploaded to GEO with identifier GSE167956 [<https://www.ncbi.nlm.nih.gov/geo/query/acc.cgi?acc=GSE167956>] and is available for analysis by any interested parties.

MINOR COMMENTS

1. *“recent studies indicate that for 30% of all protein-coding genes, and 70% of those induced by a physiological stimulus such as heat shock or hypoxia, Pol II binds to the promoter and initiates transcription, but pauses 20-100 nucleotides downstream of the transcription start site, as a result of its interaction with negative elongation factor (NELF) and DRB-sensitivity inducing factor (DSIF) [16-18].”*

This should clarify that pausing occurs at ALL protein-coding genes, while being a rate-limiting step at 40-70%, depending on cell type/method), especially stimulus-responsive genes such as those induced by heatshock or hypoxia. Estimates of pausing location have varied but the latest consensus (and the cited reference) is ~ 30-60 nt downstream of the TSS.

Response: We have revised the text in the manuscript as requested on page 4 to read as follows: “...recent studies indicate that Pol II binds to the promoter and initiates transcription, but pauses 30-60 nucleotides downstream of the transcription start site, as a result of its interaction with negative elongation factor (NELF) and DRB-sensitivity inducing factor (DSIF). This pausing is the rate-limiting step in 40-70% of genes induced by a physiological stimulus such as heat shock or hypoxia [16-18].”

2. *“positive transcription elongation factor b (P-TEFb), consisting of CDK9 and cyclin T1, is recruited to paused Pol II” Would be helpful to add “through a variety of mechanisms” here plus references.*

Response: We have revised the text on page 4 to read as follows: “In response to an inducing stimulus, positive transcription elongation factor b (P-TEFb), consisting of CDK9 and cyclin T1, is recruited to paused Pol II through a variety of mechanisms and CDK9 phosphorylates three critical targets: NELF, causing it to be released from Pol II; the SPT5 subunit of DSIF, converting it into a positive elongation factor; and serine-2 (S2) in the Pol II CTD, which allows it to engage positive elongation factors, such as SPT6, and chromatin-modifying enzymes, such as SETD2, which generates the H3K36me3 mark that is associated with productive transcriptional elongation [16-19].”

3. *In the section covering background knowledge of TRIM28 and in the discussion, it should be noted that TRIM28 can itself directly bind chromatin (Bacon et al. 2020; PMID: 32402252) and the implications of this for CDK9 recruitment and the current work should be discussed.*

Response: We state in the Discussion section on page 18 of the revised manuscript that TRIM28 “may bind directly to certain DNA sequences.”

4. *In their introduction of the RIME dataset, the authors may wish to add that Pontin and Reptin have been identified as HIF interactors in Drosophila (Dekanty et al. 2010; 20585616).*

Response: We cited the reference.

5. *A brief description of RT-qPCR normalization should be included in figure legends, rather than just methods.*

Response: We have added statements regarding qPCR data normalization in all figure legends.

6. *Arrangement of panels in Fig 3C is potentially confusing.*

Response: We have revised Fig. 3C.

7. *“also inhibited by NU7441 in a dose dependent manner (Fig. 4I).” Cannot really conclude dose-dependency with only two concentrations – suggest reword to “at two doses”*

Response: We have revised the text in the manuscript in page 12 to read: “Hypoxia-induced expression of the HIF target genes *ANGPTL4* and *PDK1*, but not *RPL13A*, was also inhibited by NU7441 at two doses (Fig. 4I)”.

8. *Knockdown with rescue experiments in Fig 5E are a great approach, but it should at least be discussed/indicated how the expression levels compare to endogenous TRIM28 expression. As shown here in only the KD background it is impossible to judge. Furthermore, in this context the statement that “TRIM28 is necessary and sufficient for activation of HIF target gene expression” is a bit strong and should be toned down.*

Response: We have revised the text on page 13 to read as follows: “We stably transfected a TRIM28-KD subclone with empty vector (EV) or vector encoding TRIM28-WT, TRIM28-S824A or TRIM28-S824D, which were expressed at similar levels with the parental cells (Fig. 5E). Remarkably, HIF target gene expression was increased at both 20% and 1% O₂ in the TRIM28-S824D subclone (Fig. 5F), demonstrating that introduction of a negative charge at residue 824 of TRIM28 (by DNA-PK-dependent serine phosphorylation or aspartate substitution) is critical for activation of HIF target gene expression”.

9. *“whereas S2 phosphorylation (S2P) triggers release” should be reworded to reflect that the CTD phosphorylations have not been formally shown to trigger different steps in the transcription cycle, but rather are highly correlated/associated with certain stages.*

Response: We have revised the text on page 14 to read: “whereas S2 phosphorylation (S2P) is associated with Pol II pause release.”

10. *“indicating that TRIM28 and DNA-PK are required for S2P of paused Pol II in response to hypoxia.” This statement is somewhat too strong given that ChIP does not specifically measure transcriptional activity. Suggest rewording along the lines of “is consistent with...”*

Response: We have revised the text in the manuscript on page 14 to read as follows: “...indicating that TRIM28 and DNA-PK are required for phosphorylation of paused Pol II at Ser2 in response to hypoxia. The loss of hypoxia-induced HRE occupancy by Pol II-S2P, but not by total Pol II or Pol II-S5P in the TRIM28-KD and DNA-PKcs-KD subclones (Fig. 6A) strongly support the hypothesis that TRIM28 and DNA-PK promote the release of paused Pol II.”

11. *“subsequent productive transcriptional elongation require the phosphorylation and dissociation of NELF” should replace with “requires”*

Response: We have revised the text on page 14 to read as follows: “NELF, which is a complex containing five subunits, binds to Pol II to induce pausing, and Pol II release; subsequent productive transcriptional elongation requires the phosphorylation and dissociation of NELF [16, 17].”

12. *In Fig 6C how are the relative levels at each region measured/presented? It seems odd that the levels at the 5' region should be so much lower than the downstream regions. Please also clarify if the cDNA protocol used oligo dT and/or random hexameric primers as this may help with interpretation.*

Response: We analyzed transcriptional elongation of ANGPTL4 mRNA by RT-qPCR assay using primers designed to amplify sequences at the 5' end (within 60 nucleotides of the transcription start site), middle, and 3' end of the mRNA. For each primer pair, the qPCR data are normalized to the mean value for the NTC subclone at 20% O₂, which we now state explicitly in the Fig. 6C figure legend. To improve clarity, we have revised the Results section on page 15 to read as follows: “For each primer pair, the qPCR data are normalized to the mean value for the NTC subclone at 20% O₂. The experiment was designed to analyze the effect of hypoxia and DNA-PK or TRIM28 knockdown on transcript abundance. Hypoxia induced only a modest increase in the abundance of transcripts containing sequences at the 5' end of the mRNA (Fig. 6C), which supports the model that transcription initiation occurs regardless of O₂ concentration [13, 26]. In contrast, the abundance of transcripts containing sequences in the middle and at the 3' end of the mRNA was markedly increased when NTC cells were subjected to hypoxia (Fig. 6C). Knockdown of TRIM28 or DNA-PKcs in SUM159 cells did not affect abundance of transcripts containing sequences at the 5' end of the mRNA but significantly decreased the abundance of transcripts containing sequences in the middle and at the 3' end of ANGPTL4 mRNA specifically under hypoxic conditions (Fig. 6C).” For cDNA synthesis, we used both random octamers and oligo dT₁₆ primers.

13. *“by contrast, Pol II pausing (Fig. 6A) and transcriptional elongation (Fig. 5F) at HIF target genes under non-hypoxic conditions were not altered in TRIM28-KD breast cancer cells.” This statement is not robustly addressed by the current analysis. Suggest tone down or rephrasing.*

Response: We have revised the Discussion section on pages 19-20 to read as follows: “by contrast, our data suggest that Pol II pausing (Fig. 6A) and transcriptional elongation (Fig. 5F) at HIF target genes are not significantly affected by TRIM28-KD under non-hypoxic conditions, from which we conclude that TRIM28 is a positive regulator in the induced state and has no role in the uninduced state.”

14. *Please check for full descriptions of all methods – for example full recipes should be given for all ChIP buffers, incubation time with antibodies is missing, and reverse cross-linking usually includes elevated temperature.*

Response: The Methods on page 22 has been revised to read as follows: “**ChIP-qPCR assays.** ChIP was performed according to EMD Millipore ChIP assay kit (catalog # 17-295) with some modifications. Briefly, MDA-MB-231 and SUM159 cells were cross-linked in 1% formaldehyde for 10 min, quenched in 0.125 M glycine for 5 min, and lysed with SDS lysis buffer (1% SDS, 10

mM EDTA, 50 mM Tris [pH 8.1]). Chromatin was sheared by sonication to between 200 and 1000 bp. Lysates were diluted 1:10 with ChIP dilution buffer (0.01% SDS, 1.1% Triton X100, 1.2 mM EDTA, 16.7 mM Tris-HCl [pH 8.1], 167 mM NaCl), precleared with salmon sperm DNA/protein A-agarose slurry (MilliporeSigma) at 4 °C for 1 h, and incubated with antibody (Supplementary Table 2) in the presence of agarose beads overnight. After sequential washes with low-salt (0.1% SDS, 1% Triton X-100, 2 mM EDTA, 20 mM Tris-HCl [pH 8.1], 150 mM NaCl), high-salt (0.1% SDS, 1% Triton X-100, 2 mM EDTA, 20 mM Tris-HCl [pH 8.1], 500 mM NaCl), LiCl (0.25 M LiCl, 1% IGEPAL-CA630, 1% deoxycholic acid, 1 mM EDTA, 10 mM Tris [pH 8.1]), and Tris-EDTA buffers (10 mM Tris-HCl [pH 8.0], 1 mM EDTA), DNA was eluted in freshly prepared elution buffer (1% SDS, 0.1M NaHCO₃), and cross-links were reversed by addition of 0.2 M NaCl at 65°C for 4 h. DNA was purified by phenol-chloroform extraction and ethanol precipitation, and analyzed by qPCR. Primer sequences are shown in Supplementary Table 3.”

References

1. Shen G, Li X, Jia YF, Piazza GA, Xi Y. Hypoxia regulated microRNAs in human cancer. *Acta Pharmacologica Sinica* **34**, 336-341 (2013).
2. Sayed D, et al. MicroRNA-21 targets sprouty2 and promotes cellular outgrowths. *Mol. Biol. Cell.* **19**, 3272-3282 (2008).

Reviewer #3

This submission by Yang et al describes an interesting study of the roles of TRIM 28 and the enzyme DNA-PK in the transcriptional response to hypoxia mediated by hypoxia inducible factor. The experiments appear to have been carried out very thoroughly and explore multiple HIF family members, a range of hypoxia-inducible genes and a number of cell lines and after some minor additions should be published.

Previous studies have suggested role for programmed DNA strand breaks and the DNA damage response in transcription of heat shock, serum-induced and nuclear receptor mediated transcription (Madabhushi, R. Cell2015, 161, 1592 and Bunch et al, Nat Commun, 2015: 6, 10191). The DSB were regarded as a direct product of increased trans activation rather than an effect of the treatment used and thought to be generated by topoisomerase 2. DNA-PK was thus thought of part of the non-homologous end joining response to DSB generation during transcription. The authors here suggest a modification of this mechanism in transcriptional responses to hypoxia in which DNA-PK and its substrate TRIM28 are recruited by HIF-1 to the promoter independently of DSB. While this is an interesting and compelling mechanism, it would be good to have more evidence for the lack of involvement of topo2 and DSB in the transcriptional function of HIF1. The experiments in Fig. 4 G attempt to address this, although g-H2Ax levels are examined after eight hours hypoxia. Earlier points should be looked at as transcription is likely induced much earlier, likely within minutes. In addition, ChIP- PCR assays of g-H2Ax association with the hypoxia-induced genes during hypoxia should be carried out. It

would also be recommended to try to exclude a role for topo2 in hypoxia-inducible transcription using topo2 inhibitors.

Response: We appreciate these insightful comments from the reviewer and have performed the following experiments to address the reviewer's concerns.

First, as suggested, we exposed the SUM159 breast cancer cells to 1% O₂ for earlier time points: 0, 15, 30, or 60 minutes, and immunoblot assays were performed. The results showed hypoxia induced the signals of phosphorylation of TRIM28 at S824 and the phosphorylation of DNA-PKcs at T2609 appeared at as early as 15 minutes, along with HIF-1α protein accumulation. However, there was no evidence of phosphorylation of H2AX at S139 (γH2AX) or ATM at S1981, providing further evidence that exposure of breast cancer cells to 1% O₂ does not induce a DNA damage response. **These data are presented as Supplementary Fig. 6A in the revised manuscript.**

Second, we followed the reviewer's suggestion and performed ChIP-qPCR assay in SUM159 cells using antibody against γH2AX. The results showed that, in response to hypoxia, HIF-1α, but not γH2AX, was recruited to the HREs of the HIF target genes, *ANGPTL4* and *PDK1*. **These data are presented as Supplementary Fig. 6B in the revised manuscript.**

Finally, to investigate whether topoisomerase II is involved in hypoxia-inducible transcription of HIF target genes in breast cancer cells, we exposed SUM159 cells to 1% O₂ for 24 h in the presence of DMSO or the topoisomerase II inhibitor merbarone (Fortune and Osheroff, *J. Biol. Chem.* 1998;273:17643; Ju *et al.*, *Science* 2006;312:1798). We observed inhibition of hypoxia-induced *ANGPTL4* mRNA expression in merbarone-treated cells (panel A, below). However, we found that merbarone, as previously reported for the topoisomerase II inhibitor mitoxantrone (Toh and Li, *Clin. Cancer Res.* 2011; 17:5026), blocked hypoxia-induced HIF-1α protein stabilization (panel B, below). We also found that merbarone treatment increased the phosphorylation of TRIM28 at S824 even under non-hypoxic conditions (panel B, below). Based on these data, we conclude that further investigation of potential relationships between topoisomerase II and either HIF-1 or TRIM28 is beyond the scope of the present study.

References

1. Fortune JM, Osheroff N. Merbarone inhibits the catalytic activity of human topoisomerase II α by blocking DNA cleavage. *J. Biol. Chem.* **273**, 17643-50 (1998).
2. Ju B, et al. A topoisomerase IIb-mediated dsDNA break required for regulated transcription. *Science* **312**, 1798-1802 (2006).
3. Toh YM, Li TK. Mitoxantrone inhibits HIF-1 α expression in a topoisomerase II-independent pathway. *Clin. Cancer Res.* **17**, 5026-5037 (2011).

REVIEWER COMMENTS

Reviewer #1 (Remarks to the Author):

The authors responded my previous requests but not fully and so some critical validations are necessary to prevent potential misinterpretation and falsehood. I suggest that the authors address and revise the manuscript thoroughly according to the comments below.

1. The original question was "The authors showed that HIF1 α is not recruited to the hypoxia genes in TRIM28 KD and DNA-PK KD cells in Fig. 3E. Could it indicate that TRIM28 recruits HIF1 α to the activated genes?" The authors' response is "To address this comment, we performed ChIP-qPCR to analyze SUM159 subclones expressing a non-targeting control shRNA or shRNA targeting HIF-1 α . Hypoxia-induced recruitment of TRIM28 and DNA-PKcs was significantly decreased in cells expressing shRNA targeting HIF-1 α . These data are presented as Supplementary Fig. 2 in the revised manuscript."

How can the authors interpret these data to argue that HIF-1 α recruits TRIM28 (supported by Supplementary Fig. 2) when TRIM28 KD or DNA-PK KD blocks HIF-1 α recruitment (Fig. 3E and Supplementary Fig. 4A)? If TRIM28 KD blocks the recruitment of HIF-1 α , then it suggests that TRIM28 is required for HIF-1 α recruitment. How could the authors reconcile these two results? Are HIF-1 α and TRIM28 interdependent? How much is TRIM28 occupied at the HIF-1 α -target genes before hypoxia? In the ChIP-qPCR analysis results (e.g. Fig. 1C), what is the unit of Y-axis (currently it just says GeneX pPCR)? Is it a fold change or percent input? To this reviewer, it doesn't appear to be the percent input for the scales.. If it is a fold change, did the authors use an IgG or unoccupied locus control? In the method section, the information is not found (yet it should be) and it is unable to judge whether TRIM28 is preoccupied at the hypoxia genes before hypoxia, with the current ChIP-qPCR data presentation. It appears to me that TRIM28 is preoccupied to assist HIF-1 α recruitment upon hypoxia as Fig. 3E and Supplementary Fig. 4A in the current manuscript shows and previous findings suggest. The authors should clarify the questions above thoroughly and reinterpret/rewrite the data as necessary.

2. It was asked "In Fig. 4H, DNA-PKcs phosphorylation at T2609 was inhibited by NU7441. This residue T2609, as well as TRIM28 S824, was reported to be ATM-dependent by multiple papers (DOI: 10.1128/MCB.00048-06; DOI: 10.1042/BJ20020973), which should be stated in the manuscript. In addition, ATM inhibitor should be tested to determine whether ATM is not involved in the phosphorylation of DNA-PKcs and TRIM28. In addition to immunoblotting, ChIP-qPCR results showing DNA-PKcs T2609 and TRIM28 S824 at some hypoxia target genes are to be shown in WT and ATM- and DNA-PK-inhibited cells." In response, the authors showed that the phosphorylation of DNA-PK at T2609 is not ATM-dependent, using KU55933 and WB. In addition, the authors showed that KU55933 doesn't affect phosphorylated TRIM28 (S824) and DNA-PK (T2609) to be recruited on the two hypoxia genes by ChIP-qPCR. These two data should be included in the main figures. In addition, the normalization method and Y-axis label of the ChIP-qPCR should be validated and revised (throughout the manuscript and supplementary data for the latter). Also, again, this residue T2609, as well as TRIM28 S824, was reported to be ATM-dependent by multiple papers (DOI: 10.1128/MCB.00048-06; DOI: 10.1042/BJ20020973), which should be stated in the manuscript.

3. In page 12, it is said "Furthermore, in contrast to induction by heat shock, ATM activity is not required for HIF target gene expression in response to hypoxia." It should be rewritten by stating and citing a couple of papers suggesting the ATM requirement in the heat shock- and, in particular, hypoxia-induced gene expression.

4. In page 19-20, it is stated "our data suggest that Pol II pausing (Fig. 6A) and transcriptional elongation (Fig. 5F) at HIF target genes are not significantly affected by TRIM28-KD under non-hypoxic conditions... no major role in the uninduced state" This argument should be removed- the RT-PCR and ChIP data in Fig. 6A and 5F are insufficient to conclude about the relation between TRIM28

and Pol II pausing at hypoxia genes. If the authors want to argue that TRIM28 has no roles in Pol II pausing in hypoxia-inducible genes, Pol II ChIP-seq or RNA-seq (mapping 3' ends of RNAs) in non-hypoxic conditions should be included comparing WT and TRIM28 KD cells.

5. It is noted that the condition of IP washes is not stringent for the low salt (150 mM NaCl). It is known that TRIM28 binds to DNA-PK stably in a stringent IP condition (e.g. 0.25M KCl HEGN) with non-hypoxic nuclear extracts and so it is unexpected that TRIM28 associates with DNA-PK in a HIF-dependent manner.

6. The title of the manuscript requires the validation of Fig. 3E and Supplementary Fig. 4A vs Supplementary Fig. 2 and the ChIP-qPCR method/normalization as suggested above (Inquiry #1). It seems that there is no rational basis to argue one over the other or to choose one of the two sets of data to propose a model.

Reviewer #2 (Remarks to the Author):

Response to reviewer #2 major comment 1:

The clarification of HRE locations (is this information now included in the revised manuscript?), replicates, and data for additional genes added in Supplementary Fig. 1B go some way towards addressing the concerns. However, the issue of whether binding occurs in a defined peak-like manner vs. a broader region has not been addressed. Furthermore, the appropriateness of the HRE for assessing RNAPII CTD phosphorylation or CDK9 recruitment is still questionable given the varying locations of the HREs with respect to transcription units and the distinct locations of these factors.

Response to reviewer #2 major comment 2:

A revised description of the co-IP methodology has been included.

Response to reviewer #2 major comment 3:

A revised description of the RNA-seq methodology has been included. However, the appropriateness of the statistical comparisons has not been adequately addressed nor have the suggested minimal additional comparisons been included.

Response to reviewer #2 major comment 4:

The new volcano plot helps to address the question and the raw data have been uploaded to GEO.

Response to reviewer #2 major comment 5:

The authors confirm that at least one known hypoxia/HIF1-induced repressors, miR-21, does follow the suggested pattern. To clarify, the suggested Metascape (or similar) analysis was for the transcription factor targets analysis rather than the provided pathway analysis results. The intention was to assess if targets of known repressors are enriched among the hypoxia repressed genes, eg are known targets of miR-21 repressed?

Response to reviewer #2 minor comments:

The majority of minor comments have been addressed satisfactorily.

Response to reviewer #2 minor comment 12:

This issue has been clarified; however, it may be worth considering relabeling the y-axis in Fig. 6C as "fold-change" or similar to provide additional clarity.

Reviewer #3 (Remarks to the Author):

The authors have adequately responded to the requests in the review.

Point-by-Point Response to Reviewers' Comments

We have addressed each of the reviewers' comments in the second revision of the manuscript. Our point-by-point response to each of the reviewers' comments is provided below. Changes in the marked manuscript appear in blue font.

Reviewer #1 (Remarks to the Author):

The authors responded my previous requests but not fully and so some critical validations are necessary to prevent potential misinterpretation and falsehood. I suggest that the authors address and revise the manuscript thoroughly according to the comments below.

1. The original question was "The authors showed that HIF1 α is not recruited to the hypoxia genes in TRIM28 KD and DNA-PK KD cells in Fig. 3E. Could it indicate that TRIM28 recruits HIF1 α to the activated genes?" The authors' response is "To address this comment, we performed ChIP-qPCR to analyze SUM159 subclones expressing a non-targeting control shRNA or shRNA targeting HIF-1 α . Hypoxia-induced recruitment of TRIM28 and DNA-PKcs was significantly decreased in cells expressing shRNA targeting HIF-1 α . These data are presented as Supplementary Fig. 2 in the revised manuscript." How can the authors interpret these data to argue that HIF-1 α recruits TRIM28 (supported by Supplementary Fig. 2) when TRIM28 KD or DNA-PK KD blocks HIF-1 α recruitment (Fig. 3E and Supplementary Fig. 4A)? If TRIM28 KD blocks the recruitment of HIF-1 α , then it suggests that TRIM28 is required for HIF-1 α recruitment. How could the authors reconcile these two results? Are HIF-1 α and TRIM28 interdependent? How much is TRIM28 occupied at the HIF-1 α -target genes before hypoxia? In the ChIP-qPCR analysis results (e.g. Fig. 1C), what is the unit of Y-axis (currently it just says GeneX pPCR)? Is it a fold change or percent input? To this reviewer, it doesn't appear to be the percent input for the scales. If it is a fold change, did the authors use an IgG or unoccupied locus control? In the method section, the information is not found (yet it should be) and it is unable to judge whether TRIM28 is preoccupied at the hypoxia genes before hypoxia, with the current ChIP-qPCR data presentation. It appears to me that TRIM28 is preoccupied to assist HIF-1 α recruitment upon hypoxia as Fig. 3E and Supplementary Fig. 4A in the current manuscript shows and previous findings suggest. The authors should clarify the questions above thoroughly and reinterpret/rewrite the data as necessary.

Response: We have revised the Discussion to include the following paragraph on pages 21-22: "In this paper we demonstrate that HIFs recruit TRIM28 and DNA-PK to HREs, and that TRIM28 and DNA-PK recruitment is required for stable HIF occupancy. Many transcriptional regulatory complexes are dependent upon multiple cooperative interactions for complex formation and maintenance. As a result, DNA-binding transcription factors are required to recruit coactivator complexes to specific DNA binding sites and the coactivator complexes are in turn necessary for stable chromatin occupancy by DNA-binding transcription factors. This effect of coactivators on DNA binding is mediated either by protein-protein interactions that stabilize protein-DNA interactions or, in the case of chromatin modifying enzymes, by increasing access of the transcription factor to its DNA binding site. Examples include the following: (a) knockdown of any member of a coactivator complex containing S100A10, ANXA2, and SPT6 inhibits the binding of OCT4, as well as the remaining members of the complex, to the NANOG, SOX2, and

KLF4 genes [67]; (b) HIFs recruit peptidylarginine deiminase 4 (PADI4) to HREs in a HIF-dependent manner and PADI4 recruitment is in turn required for histone citrullination and stable occupancy of HREs by HIFs [68]; (c) HIF-1 recruits the histone demethylase JMJD2C/KDM4C to HREs and JMJD2C recruitment is in turn required for histone demethylation and stable occupancy of HREs by HIF-1 [9]; and (d) HIF-1 recruits PKM2 to HREs and PKM2 enhances chromatin occupancy of HREs by HIF-1 [69]. In all such cases, as well as the results of the current study, it is the specificity of the DNA binding transcription factor (e.g. HIF-1) that determines where (e.g. at HREs) and when (in response to hypoxia) complex formation occurs.”

In the current study, we have focused on delineating the molecular mechanism underlying the transcriptional response to hypoxia and present data as fold change under hypoxic conditions (1% O₂) as compared to non-hypoxic conditions (20% O₂). In the revised manuscript, we have relabeled the Y-axis of ChIP-qPCR bar graphs as “Fold enrichment” and described the normalization method in the methods section (page 24). Finally, it should be noted that most cancer cell lines have basal HIF-1 α , HIF-2 α and HIF-1 β expression detectable by immunoblot assays as well as basal HIF-1 and HIF-2 chromatin occupancy at HREs detectable by ChIP-qPCR assays, providing an explanation for basal TRIM28 occupancy at HREs under non-hypoxic conditions.

2. It was asked “In Fig. 4H, DNA-PKcs phosphorylation at T2609 was inhibited by NU7441. This residue T2609, as well as TRIM28 S824, was reported to be ATM-dependent by multiple papers (DOI: 10.1128/MCB.00048-06; DOI: 10.1042/BJ20020973), which should be stated in the manuscript. In addition, ATM inhibitor should be tested to determine whether ATM is not involved in the phosphorylation of DNA-PKcs and TRIM28. In addition to immunoblotting, ChIP-qPCR results showing DNA-PKcs T2609 and TRIM28 S824 at some hypoxia target genes are to be shown in WT and ATM- and DNA-PK-inhibited cells.” In response, the authors showed that the phosphorylation of DNA-PK at T2609 is not ATM-dependent, using KU55933 and WB. In addition, the authors showed that KU55933 doesn’t affect phosphorylated TRIM28 (S824) and DNA-PK (T2609) to be recruited on the two hypoxia genes by ChIP-qPCR. These two data should be included in the main figures. In addition, the normalization method and Y-axis label of the ChIP-qPCR should be validated and revised (throughout the manuscript and supplementary data for the latter). Also, again, this residue T2609, as well as TRIM28 S824, was reported to be ATM-dependent by multiple papers (DOI: 10.1128/MCB.00048-06; DOI: 10.1042/BJ20020973), which should be stated in the manuscript.

Response: As suggested, we have: (a) moved Supplementary Fig. 6D and E to Figure 5C and 5G, respectively; (b) relabeled the Y-axis of ChIP-qPCR bar graphs and described the normalization method (page 24); and (c) discussed ATM-dependent phosphorylation of DNA-PKcs at T2609 and TRIM28 at S824 on page 19 and cited relevant papers in the revised manuscript as follows: “Prior studies have reported that in response to DSBs, DNA-PKcs is phosphorylated at T2609 and TRIM28 is phosphorylated at S824, both of which are ATM-dependent and critical for their function in DNA damage pathways [56-59]. Thus, the interaction of HIF-1 α and TRIM28 appears to substitute for DNA damage as a stimulus for DNA-PK assembly and activity in response to hypoxia.”

3. In page 12, it is said “Furthermore, in contrast to induction by heat shock, ATM activity is not required for HIF target gene expression in response to hypoxia.” It should be rewritten by stating and citing a couple of papers suggesting the ATM requirement in the heat shock- and, in particular, hypoxia-induced gene expression.

Response: We have revised the statement on page 12 and cited corresponding papers in the current version of manuscript to read as follows: “These results indicate that exposure of breast cancer cells to 1% O₂ does not cause detectable DNA damage or increased ATM activity. In contrast, ATM is required for heat shock-induced gene expression and has been implicated in HIF-1 dependent hypoxia signaling [18, 19, 34, 35], although it appears that DNA damage occurs only at O₂ levels below 0.1% [36-38].”

4. In page 19-20, it is stated “our data suggest that Pol II pausing (Fig. 6A) and transcriptional elongation (Fig. 5F) at HIF target genes are not significantly affected by TRIM28-KD under non-hypoxic conditions... no major role in the uninduced state” This argument should be removed- the RT-PCR and ChIP data in Fig. 6A and 5F are insufficient to conclude about the relation between TRIM28 and Pol II pausing at hypoxia genes. If the authors want to argue that TRIM28 has no roles in Pol II pausing in hypoxia-inducible genes, Pol II ChIP-seq or RNA-seq (mapping 3' ends of RNAs) in non-hypoxic conditions should be included comparing WT and TRIM28 KD cells.

Response: We have revised the statement on pages 20-21 of the revised manuscript to specify the data used to support as conclusions as follows: “by contrast, our data suggest that at HIF target genes relief of Pol II pausing, as inferred by Pol II-S2P ChIP (Fig. 6A), and transcriptional elongation, as inferred by RT-qPCR transcript mapping (Fig. 6C) and H3K36me3 ChIP (Fig. 6D), are not significantly affected by TRIM28-KD under non-hypoxic conditions, from which we conclude that TRIM28 is an essential positive regulator in the induced state and has no major effect on HIF target gene expression in the uninduced state.”

5. It is noted that the condition of IP washes is not stringent for the low salt (150 mM NaCl). It is known that TRIM28 binds to DNA-PK stably in a stringent IP condition (e.g. 0.25M KCl HEGN) with non-hypoxic nuclear extracts and so it is unexpected that TRIM28 associates with DNA-PK in a HIF-dependent manner.

Response: First, as shown in Fig.1I, we do observe a weak interaction of TRIM28 and DNA-PK in cells exposed to 20% O₂ and this interaction signal is greatly increased in cells exposed to 1% O₂ in a HIF-dependent manner. In order to have both signals fall within the limited dynamic range of immunoblot assays, we did not overexpose the blot. Second, we used whole cell lysates rather than nuclear extracts as the IP input, and this difference may have had some effect on the IP results. It is not clear how interaction of these proteins in cells exposed to 20% O₂ makes our finding of increased interaction in cells exposed to 1% O₂ “surprising”.

6. The title of the manuscript requires the validation of Fig. 3E and Supplementary Fig. 4A vs Supplementary Fig. 2 and the ChIP-qPCR method/normalization as suggested above (Inquiry

#1). *It seems that there is no rational basis to argue one over the other or to choose one of the two sets of data to propose a model.*

Response: As mentioned above, we have relabeled all the Y-axis of ChIP-qPCR graphs and described the normalization method on page 24 in the method section. The manuscript's 67 data panels provide more than adequate experimental support for the conclusion stated in the title.

Reviewer #2 (Remarks to the Author):

1. Response to reviewer #2 major comment 1:

The clarification of HRE locations (is this information now included in the revised manuscript?), replicates, and data for additional genes added in Supplementary Fig. 1B go some way towards addressing the concerns. However, the issue of whether binding occurs in a defined peak-like manner vs. a broader region has not been addressed. Furthermore, the appropriateness of the HRE for assessing RNAPII CTD phosphorylation or CDK9 recruitment is still questionable given the varying locations of the HREs with respect to transcription units and the distinct locations of these factors.

Response: As suggested, we have included the location of HREs relative to the transcription start site on page 6 of the revised manuscript: "The results showed that, similar to HIF-1 α , TRIM28 and all three subunits of DNA-PK were recruited to the HREs of two prototypical HIF target genes, *ANGPTL4* (HRE is located in the 5'-flanking region, 2012 base pairs upstream of the transcription start site) [24] and *PDK1* (HRE is located in the first intron, 760 base pairs downstream of the transcription start site) [25], in response to hypoxia, whereas no recruitment was observed at the *RPL13A* gene, which is not hypoxia-induced or HIF-regulated (Fig. 1C)".

Regarding the "appropriateness" of the assays, we have inserted the following statement on page 14 of the revised manuscript: "Several studies have demonstrated hypoxia-induced RNA Pol II CTD phosphorylation or CDK9 recruitment to HREs [11, 42, 43]."

2. Response to reviewer #2 major comment 2:

A revised description of the co-IP methodology has been included.

Response: Thank you.

3. Response to reviewer #2 major comment 3:

A revised description of the RNA-seq methodology has been included. However, the appropriateness of the statistical comparisons has not been adequately addressed nor have the suggested minimal additional comparisons been included.

Response: We added the following additional details regarding statistical analysis of the data to the description of RNA-seq methods on page 26 of the revised manuscript: "Differential expression results (1% vs 20% O₂) with FDR < 0.05 and mRNA fold change (FC) > 1.5 were used

as cutoff for identification of hypoxia-induced and hypoxia-repressed genes. We also compared different subclones (KD vs NTC) under hypoxic conditions to identify significantly changed mRNA expression ($FC > 1$ and $FDR < 0.05$) as HIF/DNA-PKcs/TRIM28-dependent genes. Finally, the combined list of mRNAs identified by these two strategies was used for Venn and Gene ontology analysis.”

4. *Response to reviewer #2 major comment 4:*

The new volcano plot helps to address the question and the raw data have been uploaded to GEO.

Response: Thank you.

5. *Response to reviewer #2 major comment 5:*

The authors confirm that at least one known hypoxia/HIF1-induced repressors, miR-21, does follow the suggested pattern. To clarify, the suggested Metascape (or similar) analysis was for the transcription factor targets analysis rather than the provided pathway analysis results. The intention was to assess if targets of known repressors are enriched among the hypoxia repressed genes, eg are known targets of miR-21 repressed?

Response: The demonstration that specific HIF target gene products mediate repression of specific genes will require extensive additional experimental data, including knockdown and RNAseq studies for each of the candidate negative regulators, which is beyond the scope of the current study. We have added the following statement on pages 8-9 of the revised manuscript: “BHLHE40 (also known as DEC1 or SHARP2), MXI1, TWIST1, and ZEB2 (also known as SIP1 or ZFHX1B) are transcriptional repressors that were induced by hypoxia in a HIF-dependent manner in SUM159 cells. The RNA-seq platform was not optimized for detection of microRNAs but MIR29B2CHG, MIR31HG, MIR34AHG, MIR155HG, MIR210, MIR210HG, MIR570, and MIR3125 were induced by hypoxia in a HIF-dependent manner. Further studies are required to determine the extent to which each of these repressors and miRNAs is responsible for the observed changes in mRNA expression.”

6. *Response to reviewer #2 minor comments:*

The majority of minor comments have been addressed satisfactorily.

Response: Thank you.

7. *Response to reviewer #2 minor comment 12:*

This issue has been clarified; however, it may be worth considering relabeling the y-axis in Fig. 6C as “fold-change” or similar to provide additional clarity.

Response: We have relabeled the Y-axis in Fig. 6C and Supplementary Figure 8C as “fold change” in the revised manuscript.

Reviewer #3 (Remarks to the Author):

The authors have adequately responded to the requests in the review.

Response: Thank you.

REVIEWER COMMENTS

Reviewer #1 (Remarks to the Author):

In the abstract, it says that pausing occurs at 20–100 bp downstream TSS, whereas in the introduction, 30–60 downstream TSS. These are to be revised for consistency.

In the introduction (pg 4), it says “the mechanism underlying TRIM28 recruitment and the activation of ATM...in response to heat shock...[18]” Examining the referenced paper, it showed by ChIP that total TRIM28 is not recruited or released upon heat shock but constantly present in HSP70 gene although it is phosphorylated at S824. The sentence should be revised to be consistent with the findings in the referred paper(s).

I find Fig. 5C and 5G (pg 12) before Figs. 4G-K and 5A and 5B in the main text. The figures are very disorganized, which make it difficult to follow and read the manuscript. Make it sure that all figures and their descriptions in the main text are well coordinated.

In the previous review comments:

The original question was “The authors showed that HIF1 α is not recruited to the hypoxia genes in TRIM28 KD and DNA-PK KD cells in Fig. 3E. Could it indicate that TRIM28 recruits HIF1 α to the activated genes?” The authors’ response is “To address this comment, we performed ChIP-qPCR to analyze SUM159 subclones expressing a non-targeting control shRNA or shRNA targeting HIF-1 α . Hypoxia-induced recruitment of TRIM28 and DNA-PKcs was significantly decreased in cells expressing shRNA targeting HIF-1 α . These data are presented as Supplementary Fig. 2 in the revised manuscript.” How can the authors interpret these data to argue that HIF-1 α recruits TRIM28 (supported by Supplementary Fig. 2) when TRIM28 KD or DNA-PK KD blocks HIF-1 α recruitment (Fig. 3E and Supplementary Fig. 4A)? If TRIM28 KD blocks the recruitment of HIF-1 α , then it suggests that TRIM28 is required for HIF-1 α recruitment. How could the authors reconcile these two results? Are HIF-1 α and TRIM28 interdependent? How much is TRIM28 occupied at the HIF-1 α -target genes before hypoxia? In the ChIP-qPCR analysis results (e.g. Fig. 1C), what is the unit of Y-axis (currently it just says GeneX pPCR)? Is it a fold change or percent input? To this reviewer, it doesn’t appear to be the percent input for the scales. If it is a fold change, did the authors use an IgG or unoccupied locus control? In the method section, the information is not found (yet it should be) and it is unable to judge whether TRIM28 is preoccupied at the hypoxia genes before hypoxia, with the current ChIP-qPCR data presentation. It appears to me that TRIM28 is preoccupied to assist HIF-1 α recruitment upon hypoxia as Fig. 3E and Supplementary Fig. 4A in the current manuscript shows and previous findings suggest. The authors should clarify the questions above thoroughly and reinterpret/rewrite the data as necessary.

Author Response: We have revised the Discussion to include the following paragraph on pages 21-22:

“In this paper we demonstrate that HIFs recruit TRIM28 and DNA-PK to HREs, and that TRIM28 and DNA-PK recruitment is required for stable HIF occupancy. Many transcriptional regulatory complexes are dependent upon multiple cooperative interactions for complex formation and maintenance. As a result, DNA-binding transcription factors are required to recruit coactivator complexes to specific DNA binding sites and the coactivator complexes are in turn necessary for stable chromatin occupancy by DNA-binding transcription factors. This effect of coactivators on DNA binding is mediated either by protein-protein interactions that stabilize protein-DNA interactions or, in the case of chromatin modifying enzymes, by increasing access of the transcription factor to its DNA binding site. Examples include the following: (a) knockdown of

any member of a coactivator complex containing S100A10, ANXA2, and SPT6 inhibits the binding of OCT4, as well as the remaining members of the complex, to the NANOG, SOX2, and 2

KLF4 genes [67]; (b) HIFs recruit peptidylarginine deiminase 4 (PADI4) to HREs in a HIF-dependent manner and PADI4 recruitment is in turn required for histone citrullination and stable occupancy of HREs by HIFs [68]; (c) HIF-1 recruits the histone demethylase JMJD2C/KDM4C to HREs and JMJD2C recruitment is in turn required for histone demethylation and stable occupancy of HREs by HIF-1 [9]; and (d) HIF-1 recruits PKM2 to HREs and PKM2 enhances chromatin occupancy of HREs by HIF-1 [69]. In all such cases, as well as the results of the current study, it is the specificity of the DNA binding transcription factor (e.g. HIF-1) that determines where (e.g. at HREs) and when (in response to hypoxia) complex formation occurs."

In the current study, we have focused on delineating the molecular mechanism underlying the transcriptional response to hypoxia and present data as fold change under hypoxic conditions (1% O₂) as compared to non-hypoxic conditions (20% O₂). In the revised manuscript, we have relabeled the Y-axis of ChIP-qPCR bar graphs as "Fold enrichment" and described the normalization method in the methods section (page 24). Finally, it should be noted that most cancer cell lines have basal HIF-1 α , HIF-2 α and HIF-1 β expression detectable by immunoblot assays as well as basal HIF-1 and HIF-2 chromatin occupancy at HREs detectable by ChIP-qPCR assays, providing an explanation for basal TRIM28 occupancy at HREs under non-hypoxic conditions.

 I am not convinced that TRIM28 is unbound in the examined genes with the current ChIP-qPCR data in the manuscript. They used ChIP signals in 20% as controls/normalizers (as 1), instead of IgG or a control unbound gene, to yield the fold enrichment in 1% hypoxic conditions. Therefore, it is uncertain that TRIM28 is truly absent or insignificantly bound in 20%.

The KD of TRIM28 failed the recruitment of HIFs in two figures but the authors interpret these results as TRIM28 stabilizes HIFs in HREs (pg 10), while HIF KD decreased TRIM28 occupancy in HREs, which the authors interpret that HIF recruits TRIM28 (pg 6). These double interpretations/standards should be avoided. Reorganize the figures and show the inter-dependence of TRIM28 and HIF1 recruitment to target genes upon hypoxia.

TRIM28 is a DNA binding protein with reported consensus sequences by a couple of papers and it is also reported located mostly in the TSSs of many protein-coding genes. The locations examined in the manuscript, for the two hypoxic genes ANGPTL4 and PDK1, are actually quite far (2012 and 760 bp) from the Pol II pausing site or TSSs. It is plausible that TRIM28 is in the TSS, modulating the binding of HIFs in HREs upon hypoxia. The model in Fig. 6C is not reflecting the previous findings of TRIM28 without thorough examinations of TRIM28 occupancies or with biased interpretations and might be incorrect.

In addition, Fig. 6C is strange. It doesn't tell the RNA species for Pol II pausing (5' end) because they look all the same amounts in 5', middle, and 3' end in 20% O₂. If it were for Pol II pausing, more 5' end RNAs should be detected than the middle or 3' end. In spite of that, the authors discuss Pol II pausing and pause release by this figure. In qRT-PCR, these primers for 5' End and middle cannot distinguish the full-length RNA species from shorter ones. Therefore, these qRT-PCR data merely indicate RNA transcripts/expression levels. Also, I see an increased 5' end RNAs at 20% in TRIM28 KD cells, at least 2-3 fold, compared to NTC. These might indicate the leaky transcription for impaired Pol II pausing without TRIM28. I strongly suggest reinterpretation or tone-down of TRIM28 role and occupancy in uninduced state.

Lastly, it is recommended that the title is revised to "HIF1 interacts TRIM28 and DNA-PK to release paused RNA polymerase II and activate target gene transcription in response to hypoxia."

Reviewer #2 (Remarks to the Author):

The Authors have now satisfied the vast majority of my concerns. I still have some reservations about the comprehensiveness of the ChIP analysis but recognize the large amount of additional lab-based work that would be required to fully address this. Perhaps a comment to this effect can be added in the discussion.

REVIEWER COMMENTS

Reviewer #1 (Remarks to the Author):

In the abstract, it says that pausing occurs at 20–100 bp downstream TSS, whereas in the introduction, 30–60 downstream TSS. These are to be revised for consistency.

We have revised the manuscript to read “approximately 30-60 bp downstream” in both cases (pages 2 and 4, respectively).

In the introduction (pg 4), it says “the mechanism underlying TRIM28 recruitment and the activation of ATM...in response to heat shock...[18]” Examining the referenced paper, it showed by ChIP that total TRIM28 is not recruited or released upon heat shock but constantly present in HSP70 gene although it is phosphorylated at S824. The sentence should be revised to be consistent with the findings in the referred paper(s).

We have eliminated the reference to TRIM28 recruitment. The statement on page 4 now reads: “However, the mechanisms underlying the induction of ATM and DNA-PK activity in response to heat shock remain undetermined, although it appears to involve DNA damage [19].” We have also revised the Discussion on page 20 to read: “TRIM28 is constitutively bound to the *HSPA1B* promoter...”

I find Fig. 5C and 5G (pg 12) before Figs. 4G-K and 5A and 5B in the main text. The figures are very disorganized, which make it difficult to follow and read the manuscript. Make it sure that all figures and their descriptions in the main text are well coordinated.

We have reorganized Fig. 5 so that the panels are called out in the text in alphabetical order.

In the previous review comments:

The original question was “The authors showed that HIF1 α is not recruited to the hypoxia genes in TRIM28 KD and DNA-PK KD cells in Fig. 3E. Could it indicate that TRIM28 recruits HIF1 α to the activated genes?” The authors’ response is “To address this comment, we

performed ChIP-qPCR to analyze SUM159 subclones expressing a non-targeting control shRNA or shRNA targeting HIF-1 α . Hypoxia-induced recruitment of TRIM28 and DNA-PKcs was significantly decreased in cells expressing shRNA targeting HIF-1 α . These data are presented as Supplementary Fig. 2 in the revised manuscript.” How can the authors interpret these data to argue that HIF-1 α recruits TRIM28 (supported by Supplementary Fig. 2) when TRIM28 KD or DNA-PK KD blocks HIF-1 α recruitment (Fig. 3E and Supplementary Fig. 4A)? If TRIM28 KD blocks the recruitment of HIF-1 α , then it suggests that TRIM28 is required for HIF-1 α recruitment. How could the authors reconcile these two results? Are HIF-1 α and TRIM28 interdependent? How much is TRIM28 occupied at the HIF-1 α -target genes before hypoxia? In the ChIP-qPCR analysis results (e.g. Fig. 1C), what is the unit of Y-axis (currently it just says GeneX pPCR)? Is it a fold change or percent input? To this reviewer, it doesn’t appear to be the percent input for the scales. If it is a fold change, did the authors use an IgG or unoccupied locus control? In the method section, the information is not found (yet it should be) and it is unable to judge whether TRIM28 is preoccupied at the hypoxia genes before hypoxia, with the current ChIP-qPCR data presentation. It appears to me that TRIM28 is preoccupied to assist HIF-1 α recruitment upon hypoxia as Fig. 3E and Supplementary Fig. 4A in the current manuscript shows and previous findings suggest. The authors should clarify the questions above thoroughly and reinterpret/rewrite the data as necessary.

Author Response: We have revised the Discussion to include the following paragraph on pages 21-22: “In this paper we demonstrate that HIFs recruit TRIM28 and DNA-PK to HREs, and that TRIM28 and DNA-PK recruitment is required for stable HIF occupancy. Many transcriptional regulatory complexes are dependent upon multiple cooperative interactions for complex formation and maintenance. As a result, DNA-binding transcription factors are required to recruit coactivator complexes to specific DNA binding sites and the coactivator complexes are in turn necessary for stable chromatin occupancy by DNA-binding transcription factors. This effect of coactivators on DNA binding is mediated either by protein-protein interactions that stabilize protein-DNA interactions or, in the case of chromatin modifying enzymes, by increasing access of the transcription factor to its DNA binding site. Examples include the following: (a) knockdown of any member of a coactivator complex containing S100A10, ANXA2, and SPT6 inhibits the binding of OCT4, as well as the remaining members of the complex, to the NANOG, SOX2, and KLF4 genes [67]; (b) HIFs recruit peptidylarginine deiminase 4 (PADI4) to HREs in a HIF-dependent manner and PADI4 recruitment is in turn required for histone citrullination and stable occupancy of HREs by HIFs [68]; (c) HIF-1 recruits the histone demethylase JMJD2C/KDM4C to HREs and JMJD2C recruitment is in turn required for histone demethylation and stable occupancy of HREs by HIF-1 [9]; and (d) HIF-1 recruits PKM2 to HREs and PKM2 enhances chromatin occupancy of HREs by HIF-1 [69]. In all such cases, as well as the results of the current study, it is the specificity of the DNA binding transcription factor (e.g. HIF-1) that determines where (e.g. at HREs) and when (in response to hypoxia) complex formation occurs.” In the current study, we have focused on delineating the molecular mechanism underlying the transcriptional response to hypoxia and present data as fold change under hypoxic conditions (1% O₂) as compared to non-hypoxic conditions (20% O₂). In the revised manuscript, we have relabeled the Y-axis of ChIP-qPCR bar graphs as “Fold enrichment” and described the normalization method in the methods section (page 24). Finally, it should be noted that most cancer cell lines have basal HIF-1 α , HIF-2 α and HIF-1 β expression

detectable by immunoblot assays as well as basal HIF-1 and HIF-2 chromatin occupancy at HREs detectable by ChIP-qPCR assays, providing an explanation for basal TRIM28 occupancy at HREs under non-hypoxic conditions.

 I am not convinced that TRIM28 is unbound in the examined genes with the current ChIP-qPCR data in the manuscript. They used ChIP signals in 20% as controls/normalizers (as 1), instead of IgG or a control unbound gene, to yield the fold enrichment in 1% hypoxic conditions. Therefore, it is uncertain that TRIM28 is truly absent or insignificantly bound in 20%.

We do not state anywhere in the manuscript that “TRIM28 is truly absent or insignificantly bound.”

The KD of TRIM28 failed the recruitment of HIFs in two figures but the authors interpret these results as TRIM28 stabilizes HIFs in HREs (pg 10), while HIF KD decreased TRIM28 occupancy in HREs, which the authors interpret that HIF recruits TRIM28 (pg 6). These double interpretations/standards should be avoided. Reorganize the figures and show the inter-dependence of TRIM28 and HIF1 recruitment to target genes upon hypoxia.

We have revised page 10 by changing “recruitment of TRIM28 and DNA-PKcs” to “interaction with TRIM28 and DNA-PKcs.”

TRIM28 is a DNA binding protein with reported consensus sequences by a couple of papers and it is also reported located mostly in the TSSs of many protein-coding genes. The locations examined in the manuscript, for the two hypoxic genes ANGPTL4 and PDK1, are actually quite far (2012 and 760 bp) from the Pol II pausing site or TSSs. It is plausible that TRIM28 is in the TSS, modulating the binding of HIFs in HREs upon hypoxia. The model in Fig. 6C is not reflecting the previous findings of TRIM28 without thorough examinations of TRIM28 occupancies or with biased interpretations and might be incorrect.

We have added the following statement to the Discussion section on page 18: “Further studies are required to determine whether TRIM28 also binds directly to DNA at HIF target genes as previously reported for the *HSPA1B* gene [18].”

In addition, Fig. 6C is strange. It doesn't tell the RNA species for Pol II pausing (5' end) because they look all the same amounts in 5', middle, and 3' end in 20% O₂. If it were for Pol II pausing, more 5' end RNAs should be detected than the middle or 3' end. In spite of that, the authors discuss Pol II pausing and pause release by this figure. In qRT-PCR, these primers for 5' End and middle cannot distinguish the full-length RNA species from shorter ones. Therefore, these qRT-PCR data merely indicate RNA transcripts/expression levels. Also, I see an increased 5' end RNAs at 20% in TRIM28 KD cells, at least 2–3 fold, compared to NTC. These might indicate the leaky transcription for impaired Pol II pausing without TRIM28. I strongly suggest reinterpretation or tone-down of TRIM28 role and occupancy in uninduced state.

Unfortunately, despite our previous revision, the reviewer still does not seem to understand the experimental design. We have added to our already detailed description of the experiment on page

15 the following additional statement: “The experimental design does not allow comparison of the relative levels of the three different qPCR products because of potential differences in primer efficiency.”

Lastly, it is recommended that the title is revised to “HIF1 interacts TRIM28 and DNA-PK to release paused RNA polymerase II and activate target gene transcription in response to hypoxia.”

We have made a more grammatically correct revision to satisfy the reviewer’s demand to remove the word “recruitment” from the title and elsewhere (pages 5 and 10).

Reviewer #2 (Remarks to the Author):

The Authors have now satisfied the vast majority of my concerns. I still have some reservations about the comprehensiveness of the ChIP analysis but recognize the large amount of additional lab-based work that would be required to fully address this. Perhaps a comment to this effect can be added in the discussion.

We have added the following statement to the Discussion section on pages 18-19: “We have demonstrated hypoxia-induced recruitment of TRIM28 and DNA-PK to 10 out of 10 HIF target genes analyzed by ChIP-qPCR and demonstrated by RNA-seq that the vast majority of HIF target genes require TRIM28 and DNA-PK for hypoxia-induced expression. A more comprehensive analysis will require using ChIP-seq to compare the genomic footprints of TRIM28, DNA-PK, HIF-1, and HIF-2.”